# RPM: Reasoning-Level Personalization for Black-Box Large Language Models

**Jieyong Kim**[1]* **Tongyoung Kim**[1]* **Soojin Yoon**[1] **Jaehyung Kim**[1] **Dongha Lee**[1]†

[1] Department of Artificial Intelligence, Yonsei University

`{jieyong99,dykim,soojiny,jaehyungk,donalee}@yonsei.ac.kr`

## Abstract

While black-box large language models are widely deployed, they produce generic outputs that overlook individual user preferences. Current personalization methods are fundamentally limited to response-level personalization; they only match final outputs, failing to model the underlying reasoning that connects user behavior to responses. To address this, this work introduces reasoning-level personalization as a new paradigm and proposes RPM, the first systematic framework that automatically discovers user-specific reasoning structures from raw behavioral data to guide the model's personalized inference. RPM constructs a structured model of user behavior—built from response-influential features and statistical factors—to create personalized reasoning paths and retrieve beneficial examples for guiding inference through a feature-based retrieval mechanism. Extensive experiments across four diverse tasks demonstrate that RPM consistently outperforms existing response-level methods while simultaneously enhancing both personalization performance and interpretability, providing a promising direction for black-box LLM personalization. Our code is publicly available at https://github.com/jieyong99/RPM.

## 1 Introduction

Recent advances in large language models (LLMs) have significantly improved performance across a wide range of natural language processing tasks (Hendrycks et al., 2020; Liu et al., 2023; Dai et al., 2023; Liu et al., 2024). Most deployed models operate as black-box systems where internal parameters are inaccessible (Brown et al., 2020; Achiam et al., 2023; Team et al., 2024), posing a fundamental challenge to providing personalized responses that align with individual user preferences and behavioral patterns. This has led to growing interest in black-box LLM personalization (Zhang et al., 2024; Kirk et al., 2023; Kim & Yang, 2024; Zhuang et al., 2024), which aims to tailor model outputs to user-specific contexts without modifying the model parameters.

Current black-box LLM personalization approaches fall into two main categories. Retrieval-based methods, which select historical data via similarity (Salemi et al., 2023) or utility scoring (Zhuang et al., 2024), and prompting-based methods, which refine inputs through heuristic templates (Salemi et al., 2023) or iterative updates (Kim & Yang, 2024). However, both strategies share a fundamental limitation: **they focus exclusively on Response-Level Personalization, where the objective is limited to matching the final output** (Figure 1, Top). This limitation creates two key challenges. First, **Superficial Pattern Learning**. These systems can only learn shallow correlations between the overall input and the final output, failing to capture how specific components within the input influence the response. Second, **Lack of Interpretability**. Without an explicit reasoning path, it is hard to determine whether the model's output reflects authentic user preferences or misleading correlations, which threatens the reliability of the system.

In response to these limitations, we propose **Reasoning-Level Personalization as a new paradigm that aims to model the reasoning process inferred from user behaviors**. While this paradigm holds the potential for deeper behavioral insight and interpretability, realizing these advantages is a significant challenge. Our experiments confirm this: the most straightforward approach, applying

---

*Equal contribution.
†Corresponding Author

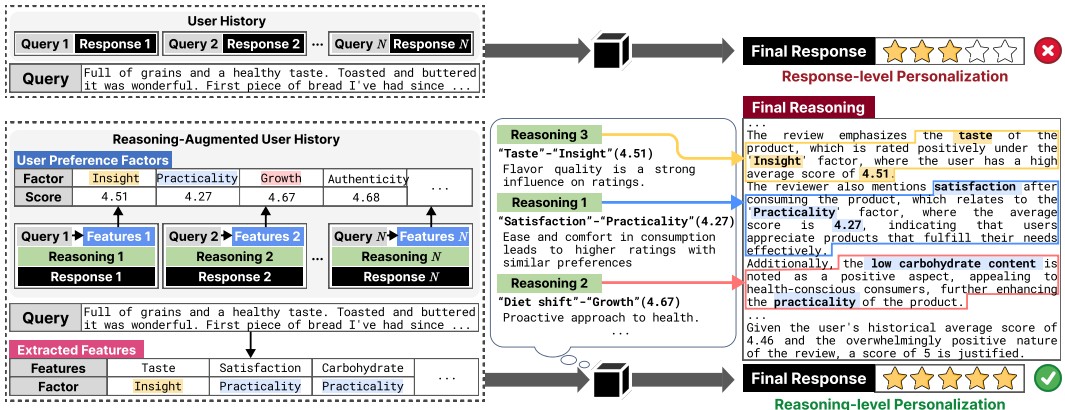

Figure 1: Comparison of response-level (Top) and reasoning-level (Bottom) personalization in a rating prediction task with scores from 1 to 5. Our approach generates personalized reasoning paths based on user-specific factors, enabling more accurate and interpretable predictions.

zero-shot chain-of-thought (CoT) prompting (Kojima et al., 2022), yields inconsistent performance compared to zero-shot baselines. A more advanced approach, constructing reasoning paths for relevant historical data as few-shot CoT examples, also fails to improve upon few-shot baselines. These failures highlight a critical gap: the absence of a systematic framework capable of transforming raw behavioral patterns into a structured reasoning model that the LLM can reliably follow.

To address this critical need, this paper introduces RPM, **R**easoning-Level **P**ersonalization for Black-Box LL**M**, a novel framework that aligns an LLM's reasoning process with a user's behavior by automatically discovering user-specific reasoning structures from raw behavioral data and leveraging them as personalized reasoning paths during inference (Figure 1, Bottom). Unlike generic structured reasoning methods (e.g., CoT, ToT) that apply task-agnostic logic regardless of who the user is, RPM derives its reasoning structure directly from each user's observed behavior, making it a data-driven modeling principle rather than a prompting strategy.

This is achieved through four key innovations: (1) Instead of naively summarizing user history into a profile (Richardson et al., 2023; Kim et al., 2024b; Sun et al., 2024; Kim et al., 2025b), the framework constructs a structured user model by extracting response-influential features from each interaction, grouping them into quantifiable factors, and assigning statistical meaning. (2) It then builds personalized reasoning based on this structured model. Providing these reasoning-augmented examples during inference effectively aligns the LLM's logic with the user's, leading to significant performance gains. (3) This strong alignment is further enhanced by feature-based retrieval, a mechanism that effectively retrieves beneficial samples based on features, providing a more relevant foundation for reasoning than simple topical matching. (4) Finally, the framework achieves high interpretability by grounding its outputs in structured reasoning paths, making individual components like features and factors explicit in the final reasoning process.

In summary, our contributions are threefold:

- The introduction and formalization of reasoning-level personalization, a new paradigm that shifts the focus from matching final responses to aligning the underlying reasoning process of LLMs with a user's specific behavior pattern.

- The proposal of RPM, a data-driven framework that automatically discovers user-specific reasoning structures from raw behavioral data and guides an LLM's inference with high interpretability in black-box LLM personalization.

- Comprehensive empirical evidence validating the framework's effectiveness. The experiments demonstrate state-of-the-art performance across four diverse tasks, confirm the significant contribution of each core technical component, and verify cross-model transferability across multiple backbone LLMs. Furthermore, human evaluations verify that RPM 's outputs are significantly more interpretable and trustworthy.

## 2 RELATED WORKS

### 2.1 RETRIEVAL-BASED LLM PERSONALIZATION

A common strategy for black-box LLM personalization is to leverage a few pieces of past user information to condition model behavior during inference. In-Context Learning (ICL) achieves this by inserting several raw query–response pairs from a user's history into the prompt (Dai et al., 2023; Liu et al., 2023; Kang et al., 2023; Di Palma et al., 2023; Wang & Lim, 2023), relying on the model's in-context generalization ability to reproduce personalized responses. Retrieval-Augmented Generation (RAG) improves this approach by retrieving semantically similar examples from a longer history (Salemi et al., 2023; 2024; Gao et al., 2023; Li et al., 2023), enabling scalability and better contextual fit. These approaches depend on query-level signals to retrieve user data, often including examples that appear relevant but offer little guidance on how to solve the current query. HYDRA (Zhuang et al., 2024) addresses this issue by training a reranker that reranks the retrieved examples based on usefulness rather than similarity. However, the utility criteria require additional model parameter training, which increases complexity. In contrast, our method retrieves examples based on structured features that serve as the foundation for reasoning, offering an effective criterion for retrieving useful reasoning examples without requiring additional model training.

### 2.2 PROMPT-OPTIMIZED LLM PERSONALIZATION

Another approach to black-box LLM personalization involves refining prompts or profiles to better align model outputs with user preferences. Prompt engineering methods heuristically encode user information into prompts (Salemi et al., 2023; Hwang et al., 2023), while Fermi improves prompt quality via iterative updates based on model feedback (Kim & Yang, 2024). Profile-based methods further summarize user traits into natural language descriptions appended to the prompt (Richardson et al., 2023; Kim et al., 2024b; 2025b; Kim et al.). However, even when effective prompts are found, these methods provide limited guidance on how to adaptively utilize input information in a case-specific way for each new query. Furthermore, it remains unclear how the provided input is utilized in generating the output, as these methods lack an explicit mapping between input attributes and reasoning steps. In contrast, our method conditions the model on examples containing user-specific reasoning paths constructed from user history, enabling reasoning-level personalization grounded in personal behavior patterns. Through structured features and factors, our method produces interpretable responses that reveal how specific query components contribute to the model's predictions.

## 3 RPM: REASONING-LEVEL PERSONALIZATION FOR BLACK-BOX LLMS

In this section, we present RPM, a framework for reasoning-level personalization of black-box LLMs using structured user information. RPM aligns model inference with user-specific behavior patterns by constructing and leveraging personalized reasoning paths from history. The framework consists of three key components: (1) *personalized factor construction*, which extracts and groups response-relevant features into statistical user-level factors (Section 3.2); (2) *personalized reasoning construction*, which builds personalized reasoning paths based on user's past responses (Section 3.3); and (3) *reasoning-aligned generation*, which retrieves and applies these reasoning paths to generate accurate and interpretable outputs (Section 3.4). Figure 2 shows the overview of our RPM framework. For details on the notation and algorithms, please refer to Appendix A and Appendix B.

### 3.1 PRELIMINARIES

Black-box LLM personalization aims to align the output of a Black-Box LLM $\mathcal{M}$ with user preferences, based on their history, without access to model parameters. We consider a set of users $\mathcal{U}$, where each user $u \in \mathcal{U}$ is associated with a history $\mathcal{H}_u = \{(q_i, a_i)\}_{i=1}^{N}$ of $N$ query-response pairs, with $q_i$ as a query and $a_i$ as the corresponding response. This history captures the user's behavioral tendencies and underlying personal logic that influence response generation. At inference time, given a target query $q'$ for user $u$, the model generates a personalized output $a' = \mathcal{M}(q', c_u(q'))$. Here, $c_u(q')$ denotes a user-specific query context derived from $\mathcal{H}_u$ via strategies such as retrieval-based selection, prompt construction, or other conditioning mechanisms.

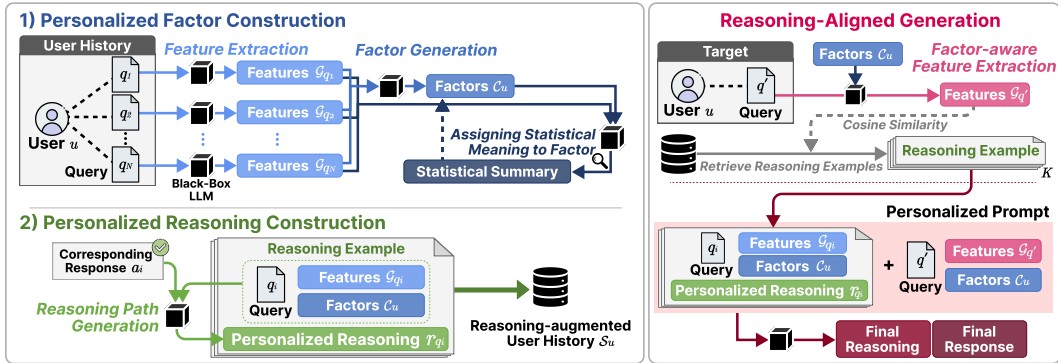

Figure 2: Overview of RPM. It extracts user-specific features/factors from user history and constructs reasoning examples by annotating personalized reasoning paths for query-response pairs. At inference time, it retrieves examples and generates the reasoning-aligned output guided by them.

## 3.2 PERSONALIZED FACTOR CONSTRUCTION

To structurally organize user information and support reasoning-level personalization, we extract structured *features* from each query $q_i$ and aggregate them into user-level semantic clusters, denoted as *factors*, which summarize user-level characteristics.

**Structured feature extraction.** Given a user's history $\mathcal{H}_u$, we extract features from each query $q_i$ to identify elements that may influence the user's response $a_i$. A raw query may contain elements that are irrelevant to the prediction, introducing noise into LLM personalization. To address this, we utilize $\mathcal{M}$ to extract features from each query $q_i$ that are likely to influence the user's response, leveraging the LLM's ability to capture semantically relevant signals from text (Kim et al., 2024a; Seo et al., 2024). Specifically, we prompt the LLM with the feature extraction instruction for the query $q_i$, resulting in a set of features $\mathcal{G}_{q_i} = \{f_j\}_{j=1}^{|\mathcal{G}_{q_i}|}$ defined as follows: $\mathcal{G}_{q_i} = \mathcal{M}(q_i)$, $\quad f_j = (\texttt{name}_j, \texttt{context}_j, \texttt{factor}_j)$, where $f_j$ denotes a $j$-th feature extracted from $q_i$, consisting of three components: $\texttt{name}_j$ is a concise semantic label that identifies the feature (e.g., "Taste"); $\texttt{context}_j$ is a disambiguating phrase that grounds the feature in its surrounding query, clarifying how it should be interpreted (e.g., "the flavor of the product"); and $\texttt{factor}_j$ is a placeholder for the higher-level semantic cluster to which the feature will be assigned during the subsequent factor generation step described below. This process yields a query-specific feature set $\mathcal{G}_{q_i}$ by identifying potentially influential elements from $q_i$ that are explicitly recognizable to the language model. These extracted features not only reduce the noise and ambiguity of the raw query, but also serve as foundations for supporting both downstream factor construction and reasoning over user's underlying patterns. Additional experiments on feature composition are provided in the Appendix E.2.

**Factor generation via LLM-based clustering.** To capture generalized user-level behavior patterns, we group the extracted features across all queries $\{q_i\}_{i=1}^N$ into a set of factors, which are higher-level semantic clusters. We adopt the LLM-based clustering method proposed in Wang et al. (2023), which groups features based on goal-related semantic similarity. This method aligns well with our objective, as semantically coherent clusters often reflect consistent reasoning tendencies unique to each user. Formally, the clustering process produces a set of user-specific factors: $\{F^{(m)}\}_{m=1}^M = \texttt{LLM\_Cluster}\left(\bigcup_{i=1}^N \mathcal{G}_{q_i}\right)$. Here, each factor $F^{(m)} = \{f_j\}_{j=1}^{|F^{(m)}|}$ represents a semantically coherent subset of features identified by the clustering algorithm. This LLM-driven semantic clustering transforms low-level, query-specific features into structured, user-level representations that not only capture recurring semantic patterns but also serve as containers for storing quantitatively analyzable statistics. A more detailed explanation about clustering is in Appendix C.

**Assigning statistical meaning to factors.** Given the set of factors, we enrich each factor with statistical summaries $\theta^{(m)}$ derived from the user's response behavior, forming the user-level factor representation $\mathcal{C}_u = \{F^{(m)}, \theta^{(m)}\}_{m=1}^M$. These summaries capture high-level user characteristics and serve as reference points during inference. First, for tasks where the response space is defined over discrete classes $y \in \mathcal{Y}$ (e.g., rating prediction), we define the *propensity* score of each response

conditioned on the presence of the factor's features:

$$\texttt{Propensity}(y, F^{(m)}) = \frac{\sum_{(q_i,a_i)\in\mathcal{H}_u} \mathbb{I}[a_i = y \land F^{(m)} \cap \mathcal{G}_{q_i} \neq \emptyset]}{\sum_{(q_i,a_i)\in\mathcal{H}_u} \mathbb{I}[F^{(m)} \cap \mathcal{G}_{q_i} \neq \emptyset]}, \quad (1)$$

where $\mathbb{I}[\cdot]$ denotes the indicator function and $\mathcal{G}_{q_i}$ is the feature set extracted from query $q_i$.

For tasks without predefined response classes or with open-ended responses, deriving direct statistical signals is challenging. Following established principles in preference modeling and explainable recommendation (Zhang et al., 2014; Cheng et al., 2023; Zhang & Chen, 2020), we characterize each factor along three complementary dimensions—frequency of appearance (coverage), degree of contribution to the response (influence), and directionality of that contribution (polarity).

As a straightforward approach, we employ $\mathcal{M}$ as an LLM evaluator to determine, for each feature, whether it influenced the final response and, if so, in which direction (i.e., polarity). This allows us to construct quantitative statistics by leveraging the fact that each factor encapsulates a group of response-influential features, even in tasks without discrete response classes. Given a query $q_i$, response $a_i$, and associated feature set $\mathcal{G}_{q_i}$, we prompt $\mathcal{M}$ to evaluate each feature $f_j \in \mathcal{G}_{q_i}$. For each feature, $\mathcal{M}$ returns whether it influenced the final response ($\texttt{IsInfl}_{f_j \to a_i} \in \{\texttt{True}, \texttt{False}\}$), and, if so, its polarity label ($\texttt{Eval}_{f_j \to a_i} \in \{\texttt{Pos}, \texttt{Neu}, \texttt{Neg}\}$).

To be specific, we first measure the *coverage* of a factor $F^{(m)}$ by counting the number of user information in which any feature from $F^{(m)}$ appears:

$$\texttt{Coverage}(F^{(m)}) = \sum_{(q_i,a_i)\in\mathcal{H}_u} \mathbb{I}[\exists f_j \in F^{(m)} \cap \mathcal{G}_{q_i}]. \quad (2)$$

Among the covered instances, we compute the number of cases where at least one feature in $F^{(m)}$ is judged to have influenced the response, yielding the *influence* count:

$$\texttt{Influence}(F^{(m)}) = \sum_{(q_i,a_i)\in\mathcal{H}_u} \mathbb{I}[\exists f_j \in F^{(m)} \cap \mathcal{G}_{q_i} : \texttt{IsInfl}_{f_j \to a_i} = \texttt{True}]. \quad (3)$$

Finally, we define the polarity score for factor $F^{(m)}$ by counting the polarity labels of all features within $F^{(m)}$ that were marked as influenced. For a polarity category $e$,

$$\texttt{Polarity}(e, F^{(m)}) = \frac{\sum_{(q_i,a_i)\in\mathcal{H}_u} \sum_{f_j\in F^{(m)}\cap\mathcal{G}_{q_i}} \mathbb{I}[\texttt{IsInfl}_{f_j \to a_i} = \texttt{True} \land \texttt{Eval}_{f_j \to a_i} = e]}{\sum_{(q_i,a_i)\in\mathcal{H}_u} \sum_{f_j\in F^{(m)}\cap\mathcal{G}_{q_i}} \mathbb{I}[\texttt{IsInfl}_{f_j \to a_i} = \texttt{True}]}. \quad (4)$$

The resulting metric values computed in Equations (1)–(4) are stored as $\theta^{(m)}$. Together, these statistics provide a quantifiable profile for each factor, characterizing how frequently it appears, how often it meaningfully contributes to predictions, and in what direction it tends to influence the response. The validity of the extracted features and aggregated factors is evaluated in Section 4.4.

## 3.3 PERSONALIZED REASONING CONSTRUCTION

Once features and factors are extracted from user history, we generate a personalized inference path for each query-response pair to capture how the given information leads to the observed response in an interpretable manner. This process constructs a stable behavioral rationale from observed responses rather than reconstructing internal cognition. While the extracted features and aggregated factors provide a rich representation of user-specific information, they do not explicitly indicate how these elements influence the final prediction. To address this, we prompt an LLM $\mathcal{M}$ with the reasoning instruction, query $q_i$, its associated features $\mathcal{G}_{q_i}$, user-level factors $\mathcal{C}_u$, and the corresponding response $a_i$, and instruct it to generate a reasoning path that connects the relevant information to the response: $r_{q_i} = \mathcal{M}(q_i, \mathcal{G}_{q_i}, \mathcal{C}_u, a_i)$ where $r_{q_i}$ denotes the generated reasoning path that explains the user's behavioral pattern based on provided query elements. Each reasoning path is then stored in the user's memory $\mathcal{S}_u$, referred to as *reasoning-augmented user history*, as part of a tuple containing all relevant components: $\mathcal{S}_u = \{(q_i, \mathcal{G}_{q_i}, r_{q_i}, a_i) | (q_i, a_i) \in \mathcal{H}_u\}$. This reasoning augmentation allows RPM to retain query-level reasoning behavior, enabling inference to be guided by personalized examples that encapsulate structural information and user-aligned behavior patterns.

### 3.4 Reasoning-Aligned Generation

At inference time, our goal is to generate reasoning-aligned outputs that are both personalized and interpretable by leveraging the structured features, factors and reasoning paths stored in $\mathcal{S}_u$.

**Factor-aware feature extraction.** Following the procedure described in Section 3.2, we extract features from the target query $q'$ with reference to the user-specific factor set $\mathcal{C}_u$. Each extracted feature contains an associated factor field, which allows the model to leverage the corresponding statistical summaries during generation, enabling reasoning grounded in personal behavior patterns. This design grounds the feature extraction in the user's established behavioral structure, reducing noise from irrelevant query elements and linking each feature to quantifiable user-level statistics.

**Retrieving useful reasoning examples.** We retrieve reasoning examples from $\mathcal{S}_u$ that are useful for reasoning-level personalization. Our retrieval process formulates retrieval criterion by using $\mathcal{G}_{q'}$ that includes response-influential elements. This feature-based formulation provides an effective criterion for selecting personalized reasoning examples. Specifically, we compute the semantic similarity between $\mathcal{G}_{q'}$ and each stored $\mathcal{G}_q$, and retrieve the top-$K$ most relevant examples: $\mathcal{S}_{u,q'}^{\mathrm{ret}} = \{(q, \mathcal{G}_q, r_q, a) \in \mathcal{S}_u | \text{Top-K} \cos(f(\mathcal{G}_{q'}), f(\mathcal{G}_q))\}$ (5), where $f(\mathcal{G}_q)$ is the embedding of the concatenated feature texts and $\cos(\cdot)$ is cosine similarity. Each retrieved example $(q, \mathcal{G}_q, r_q, a)$ provides structured logics on how similar user-specific information was processed in the past. By formulating retrieval over features rather than raw queries, the retrieval criterion captures similarity in reasoning-relevant structure rather than surface-level topical overlap, yielding examples that are more directly useful for guiding the model's reasoning process.

**Reasoning example-augmented generation.** We guide the black-box LLM $\mathcal{M}$ using the retrieved examples. Standard few-shot prompting often provides relevant examples but lacks guidance on how the given information should be interpreted to reach the correct response. In contrast, our approach supplies not only structured features and factors but also personalized reasoning paths, illustrating how such information has been used in prior response generation. We perform inference by prompting $\mathcal{M}$ on the target query $q'$, its extracted representations $\mathcal{G}_{q'}$, user-specific factor $\mathcal{C}_u$, and the retrieved reasoning examples $\mathcal{S}_{u,q'}^{\mathrm{ret}}$, i.e., $r_{q'}, a' = \mathcal{M}(q', \mathcal{G}_{q'}, \mathcal{C}_u, \mathcal{S}_{u,q'}^{\mathrm{ret}})$. By incorporating the personalized reasoning examples, the model better interprets user-specific behavior patterns. The reasoning paths serve as explicit conditioning signals that demonstrate how structured user information was previously mapped to a response, enabling both effective personalization and transparent interpretation.

**Prompt Design.** Our prompts are based on a generalizable template. The core structure that guides the model's reasoning remains consistent across all tasks. Task-specific elements are limited to designated placeholders for task descriptions, allowing the framework to remain broadly applicable with minimal modification. Therefore, our template-based approach provides a principled and scalable strategy, enabling the framework to be generalized to new tasks with minimal, targeted adjustments. Please refer to Appendix H and I for examples.

## 4 Experiments

### 4.1 Experimental Setup

**Datasets.** We evaluate our framework on four personalization tasks: text classification, regression, text generation, and question answering. Three tasks are from LaMP (Salemi et al., 2023)—LaMP-2 (movie tagging), LaMP-3 (product rating), and LaMP-5 (paper title generation)—each subsampled with 50, 100, and 100 users, respectively, from the time-based validation splits. User histories are split chronologically into training and test sets (9:1). For personalized QA, we use GlobalOpinionQA (GOQA) (Durmus et al., 2023), treating each country as a user group. Following (Kim & Yang, 2024), we convert labels to a single answer using the highest-probability option, keeping only instances above a 0.8 threshold, yielding 46 user groups. Further details are provided in Appendix D.1.

**Baselines.** We compare RPM against a range of black-box LLM personalization baselines. In addition to the Zero-shot setting, we include in-context learning (ICL), retrieval-augmented generation (RAG) (Salemi et al., 2023), and profile-augmented prompting (PAG) (Richardson et al., 2023), which incorporate user context into the prompt. HYDRA (Zhuang et al., 2024) is a plug-and-play method that adds rerankers and adapters to prioritize user-aligned content without modifying the core LLM.

Table 1: Overall performance comparison across all benchmarks. For each method, the +CoT variant denotes the use of Chain-of-Thought prompting during inference. RPM (w/o Reasoning) disables reasoning generation for the target query while keeping retrieved reasoning paths in the input context.

| Dataset | | LaMP-2 | | LaMP-3 | | LaMP-5 | | GOQA |
|---------|------|--------|------|--------|--------|--------|--------|------|
| Method | +CoT | Acc. ↑ | F1 ↑ | MAE↓ | RMSE↓ | R-1 ↑ | R-L ↑ | Acc. ↑ |
| Zero-shot | | 0.430 | 0.360 | 0.361 | 0.680 | 0.446 | 0.364 | 0.562 |
| | ✓ | 0.411 | 0.337 | 0.323 | 0.630 | 0.434 | 0.376 | 0.557 |
| ICL | | 0.495 | 0.412 | 0.333 | 0.638 | 0.455 | 0.395 | 0.695 |
| | ✓ | 0.471 | 0.374 | 0.317 | 0.625 | 0.460 | 0.405 | 0.681 |
| RAG | | 0.526 | 0.438 | 0.363 | 0.687 | 0.462 | 0.405 | 0.773 |
| | ✓ | 0.493 | 0.415 | 0.366 | 0.690 | 0.469 | 0.412 | 0.800 |
| PAG | | 0.525 | 0.444 | 0.331 | 0.662 | 0.463 | 0.404 | 0.795 |
| | ✓ | 0.513 | 0.431 | 0.339 | 0.672 | 0.464 | 0.405 | 0.820 |
| HYDRA | | 0.526 | 0.437 | 0.324 | 0.656 | 0.463 | 0.406 | 0.800 |
| | ✓ | 0.496 | 0.406 | 0.353 | 0.672 | 0.465 | 0.409 | 0.806 |
| Fermi | | 0.526 | 0.437 | 0.328 | 0.628 | 0.465 | 0.402 | 0.800 |
| | ✓ | 0.476 | 0.377 | 0.312 | 0.635 | 0.453 | 0.395 | 0.659 |
| RPM (w/o Reasoning) | | 0.510 | 0.398 | 0.305 | 0.599 | 0.466 | 0.388 | 0.820 |
| RPM | | **0.561** | **0.463** | **0.259** | **0.548** | **0.492** | **0.416** | **0.852** |

Fermi (Kim & Yang, 2024) optimizes prompts iteratively using user profiles and feedback from misaligned responses. Details of baseline implementations are available in Appendix D.2.

**Evaluation metrics.** Following (Salemi et al., 2023; Zhuang et al., 2024; Kim & Yang, 2024), accuracy (Acc.) and F1 score (F1) are used for LaMP-2, mean absolute error (MAE) and root mean squared error (RMSE) for LaMP-3, and ROUGE-1 (R-1) (Lin & Hovy, 2003), ROUGE-L (R-L) (Lin & Och, 2004)for LaMP-5. We report accuracy for the GOQA (Durmus et al., 2023).

**Implementation details.** All baselines and RPM use GPT-4o-mini (Hurst et al., 2024) as the black-box backbone. Contriever (Izacard et al., 2021) is used for few-shot retrieval with 3 examples by default, and inference is performed with temperature 0.0 for deterministic outputs. For LLM-based clustering (Wang et al., 2023), we use the default parameters without task-specific tuning. More details of implementation details are provided in Appendix D.3

## 4.2 MAIN RESULTS

Table 1 reports the performance for four personalization tasks from the LaMP and GOQA benchmarks. Compared to the Zero-shot, all the baseline methods tend to show improved performance by incorporating user-specific context through few-shot prompting, reranking, or prompt-level optimization. However, these methods focus on the response-level personalization, adapting outputs based on contextual signals without explicitly modeling LLM's reasoning process from user behavior.

To examine whether reasoning-inductive prompting can address this limitation, we evaluate the same baselines with chain-of-thought (CoT) (Wei et al., 2022; Kojima et al., 2022) reasoning prompts. While CoT is designed to guide the model through intermediate reasoning steps, our results show that it does not consistently improve performance. In several cases, it even leads to degradation, generating longer outputs that are syntactically plausible but misaligned with the user's actual decision. These inconsistencies suggest that CoT alone is insufficient, as it primarily promotes generic logical elaboration without grounding reasoning in user-specific behavior. We further examine structured reasoning baselines, including Structured Preference Induction and Tree-of-Thought (ToT)-style prompting, and find that even these approaches remain inadequate when user-specific grounding is absent(Appendix E.6). Taken together, these results indicate that meaningful personalization requires structured grounding modeled from user behavior, rather than generic reasoning strategies alone.

In contrast, RPM explicitly targets reasoning-level personalization by constructing user-specific factors from history and using them to generate and retrieve personalized reasoning paths. This

Table 2: Ablation study of RPM by adding input components to the zero-shot setting. A simple CoT reasoning example (without using features and factors) is denoted by $r_{q_i}^{\text{CoT}} = \mathcal{M}(p_{reason}, q_i; a_i)$.

| Dataset | | | LaMP-2 | | LaMP-3 | | LaMP-5 | | GOQA |
|---|---|---|---|---|---|---|---|---|---|
| Method | Input | Output | Acc. ↑ | F1 ↑ | MAE↓ | RMSE↓ | R-1 ↑ | R-L ↑ | Acc. ↑ |
| Zero-shot | $q'$ | $a'$ | 0.430 | 0.360 | 0.361 | 0.680 | 0.446 | 0.364 | 0.562 |
| Zero-shot | $q', \mathcal{G}_{q'}, \mathcal{C}_u$ | $a'$ | 0.465 | 0.370 | 0.287 | 0.576 | 0.427 | 0.347 | 0.647 |
| Few-shot | $q', \mathcal{G}_{q'}, \mathcal{C}_u, \{(q_i, a_i)\}$ | $a'$ | 0.485 | 0.392 | 0.274 | 0.565 | 0.466 | 0.389 | 0.755 |
| Few-shot | $q', \mathcal{G}_{q'}, \mathcal{C}_u, \{(q_i, \mathcal{G}_{q_i}, a_i)\}$ | $a'$ | 0.484 | 0.393 | 0.288 | 0.572 | 0.466 | 0.393 | 0.806 |
| Few-shot | $q', \{(q_i, r_i^{\text{CoT}}, a_i)\}$ | $r_{q'}, a'$ | 0.492 | 0.416 | 0.385 | 0.715 | 0.468 | 0.411 | 0.735 |
| RPM | $q', \mathcal{G}_{q'}, \mathcal{C}_u, \{(q_i, \mathcal{G}_{q_i}, r_{q_i}, a_i)\}$ | $r_{q'}, a'$ | **0.561** | **0.463** | **0.259** | **0.548** | **0.492** | **0.416** | **0.852** |

allows the model to not only produce outputs aligned with user preferences, but also to reason in ways that reflect the user's underlying behavior pattern. Across all tasks, RPM consistently outperforms response-level approaches, demonstrating that grounding inference in structured data and personal reasoning is more effective than relying on prompt-level augmentation.

We also evaluate our method without generating explicit reasoning. Notably, some results show that the model can still generate effective personalized responses, suggesting that the provided personalized reasoning examples offer implicit guidance on how to utilize the input, even without explicit reasoning. However, omitting reasoning leads to a consistent performance drop across all tasks, emphasizing the importance of explicit reasoning generation.

Beyond the main comparison, we further validate RPM's generalizability across different backbone LLMs. Additional experiments demonstrate that RPM consistently outperforms corresponding baselines across all backbones (Appendix E.5). Moreover, the personalized reasoning memory constructed with one backbone can be directly reused by other LLMs with comparable performance, confirming strong cross-model transferability. This consistency across backbones with different inductive biases suggests that the user-specific signals captured by RPM effectively override model-inherent self-bias, enabling reliable personalization regardless of the underlying LLM.

## 4.3 Ablation Studies

To evaluate the contribution of each component in our framework to personalization performance, we conduct an ablation study, as summarized in Table 2. Starting from a Zero-shot setting that uses only the target query, we incrementally incorporate user-specific query context: user-specific statistical factors, retrieved query–answer pairs, target query features, and reasoning paths. We observe a general trend of performance improvements across tasks at each stage, indicating that both static user representations and contextualized examples contribute meaningfully to personalization. Notably, while incorporating generic reasoning paths provided a performance uplift over simpler variants, this naive approach was still insufficient to outperform the main baseline methods. The greatest improvement is achieved when personalized reasoning paths are introduced, which model user-specific behavior patterns. These findings validate that effective personalization requires not only incorporating comprehensive user information, but also modeling how individual users reason.

We observe that reasoning paths constructed only from raw query–response pairs can still yield strong performance, but performance drops when explicit features and factors are absent. These observations highlight the importance of structured components, which explains why RPM achieves superior personalization. Rather than relying on prompt-level augmentation, RPM employs a systematic framework that transforms raw user history into a structured and quantifiable model of judgment. By extracting response-influential features, aggregating them into user-specific factors, and deriving their statistical significance, the framework provides an interpretable foundation for reasoning. Based on this structured model, RPM generates explicit reasoning paths and employs feature-based retrieval to align inference with user-specific behavior pattern. As confirmed by our ablation study, each stage contributes to improved accuracy, while the framework as a whole ensures robustness and interpretability beyond surface-level personalization. This suggests that features and factors not only enhance the interpretability of reasoning and facilitate more effective retrieval, but also serve as a crucial foundation and reference point for performing personalized reasoning. Additionally, these improvements are achieved without introducing any trainable components, indicating that structured context alone can effectively refine the query representation at inference time.

Table 3: Comparison of retrieval strategies using different sources and methods. Examples are retrieved from either user history ($\mathcal{H}_u$) or reasoning-augmented history ($\mathcal{S}_u$). For example retrieval, the target query $q'$ or its extracted features $\mathcal{G}_{q'}$ can serve as an input query for each retriever.

| Dataset | | | LaMP-2 | | LaMP-3 | | LaMP-5 | | GOQA |
|---|---|---|---|---|---|---|---|---|---|
| **Source** | **Retriever** | | Acc. ↑ | F1 ↑ | MAE↓ | RMSE↓ | R-1↑ | R-L↑ | Acc. ↑ |
| $\mathcal{H}_u$ | Random | $q'$ | 0.495 | 0.412 | 0.333 | **0.638** | 0.455 | 0.395 | 0.695 |
| | BM25 | $q'$ | 0.520 | 0.432 | 0.375 | 0.707 | 0.464 | 0.407 | 0.805 |
| | Contriever | $q'$ | 0.526 | 0.438 | 0.363 | 0.687 | 0.462 | 0.405 | 0.773 |
| | HYDRA-R | $q'$ | 0.521 | 0.428 | **0.329** | 0.661 | 0.468 | 0.412 | 0.790 |
| | Contriever | $\mathcal{G}_{q'}$ | **0.530** | **0.440** | 0.373 | 0.705 | **0.484** | **0.409** | **0.809** |
| $\mathcal{S}_u$ | Random | $q'$ | 0.512 | 0.405 | 0.280 | 0.563 | 0.463 | 0.387 | 0.809 |
| | BM25 | $q'$ | 0.509 | 0.429 | 0.297 | 0.584 | 0.483 | 0.410 | 0.842 |
| | Contriever | $q'$ | 0.542 | 0.459 | 0.272 | 0.565 | 0.478 | 0.400 | 0.837 |
| | Contriever | $\mathcal{G}_{q'}$ | **0.561** | **0.463** | **0.259** | **0.548** | **0.492** | **0.416** | **0.852** |

## 4.4 Evaluating Reasoning-level Interpretability

We evaluate the quality of generated reasoning through a human study conducted on Amazon Mechanical Turk (AMT)[1], comparing RPM against HYDRA and Fermi augmented with CoT prompting. We randomly sample 200 examples from datasets and ask five human annotators per example to assess the reasoning quality across six criteria: Persuasiveness, Reasonability, Faithfulness, Interpretability, Alignment, and Overall Quality. These criteria, adapted from Kim et al. (2025a), evaluate reasoning interpretability and consistency with user-specific behavior logic. Overall, RPM receives favorable evaluations across all dimensions, with particularly high scores in interpretability and alignment (Figure 3 Left). This suggests that the structured input compo-

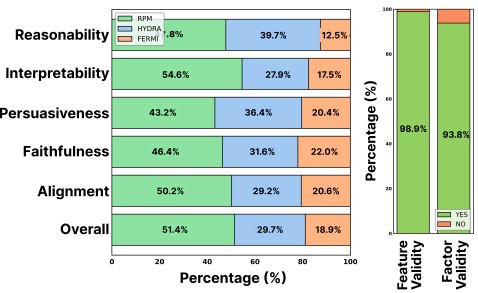

Figure 3: Human evaluation on reasoning quality and validity of feature and factor.

nents employed by RPM are clearly reflected in the model's outputs through personalized reasoning, thereby enhancing transparency and user-aligned behavior patterns. We further evaluate whether the features and factors generated by the LLM fulfill their intended roles. Results indicate that most extracted features are plausibly influential to user responses, and the constructed factors effectively cluster semantically related features coherently and meaningfully (Figure 3 Right). Additionally, we conduct a length bias analysis to ensure that the observed improvements are not driven by differences in response length (Appendix E.7). To further assess grounding reliability, we conduct a hallucination analysis on personalized reasoning paths (Appendix E.8). Details of each human evaluation protocol and case studies are provided in Appendix D.4 and Appendix I.

## 4.5 Effectiveness of the Retrieval Strategy

We investigate how different retrieval strategies affect personalization performance, focusing on both the source of examples and the retrieval method in Table 3. Overall, feature-based retrieval from the reasoning-augmented user history $S_u$ yields the most consistent and robust performance across tasks. In the upper section, methods like BM25, Contriever, and the HYDRA reranker retrieve examples based on superficial similarity to the raw query. While they show modest improvements over random selection, their effectiveness is limited, often retrieving topically relevant but decision-irrelevant examples. By contrast, the lower section uses structured retrieval over features and reasoning traces stored in $S_u$. These representations better reflect the user's choice behavior, enabling retrieval that is both contextually and logically aligned. The consistent gains highlight that retrieving from structured

---

[1]https://www.mturk.com/

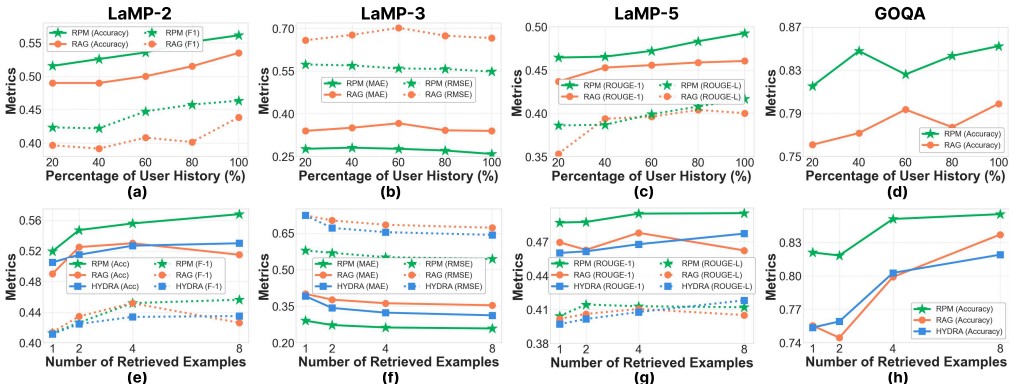

Figure 4: Performance impact of user context scale. Subfigures (a)–(d) show the effect of varying the proportion of user history, and (e)–(h) show the effect of the number of retrieved examples.

user-specific memory is far more effective than relying on raw queries. Appendix E.3 presents additional experiments on sample-level, feature-level, and factor-guided retrieval.

## 4.6 IMPACT OF USER CONTEXT SCALE ON PERSONALIZATION

**Impact of user history length.** To assess the effect of user history length on personalization performance, we vary the number of examples from user history used to construct user-specific features and factors. As shown in Figure 4, even a small number of examples enables a meaningful degree of personalization, confirming the feasibility of our approach in low-resource settings. Nevertheless, performance continues to improve with longer histories, indicating that richer context allows for more precise modeling of user behaviors and the underlying reasoning structure.

**Effect of the number of retrieved examples.** To examine the impact of the number of examples on the performance of the personalization, we vary the number of retrieved user examples provided to the model as user-specific query context. As shown in Figure 4, increasing the number of examples leads to consistent performance improvements, particularly when examples are selected based on reasoning-level similarity. This result supports our hypothesis that a richer set of aligned reasoning paths can better guide the model toward user-specific inference.

## 4.7 COST ANALYSIS

We analyze both inference latency and computational overhead on the GOQA benchmark to address scalability and efficiency. RPM requires only a minor increase in per-user inference time (0.10s vs. 0.04s for ICL/RAG) while delivering substantially higher personalization performance. Our preprocessing cost, which includes factor construction and reasoning path generation, is $0.058 per user, and per-instance inference cost is $0.0037, remaining far lower than high-overhead baselines such as Fermi ($0.32) and HYDRA ($0.47 + parameter tuning). This shows that RPM offers a practical balance, combining strong personalization and accuracy with modest overhead. For detailed results, including the token usage comparison, please refer to Appendix E.4.

## 5 CONCLUSION

We propose RPM, a novel personalization framework for black-box LLMs. RPM is designed to achieve reasoning-level personalization by aligning the model's generation process with user-specific decision pattern. Our extensive evaluations across various tasks demonstrate that RPM not only outperforms existing personalization methods, but also that each component is essential for effective personalization. Moreover, the explanations generated by RPM are interpretable, as they are explicitly grounded in structured components derived from the user's past behavior. This work represents a paradigm shift in black-box LLM personalization, moving from conventional response-level approaches to a novel reasoning-level personalization framework.

## 6 ETHICS STATEMENT

This work relies exclusively on publicly available datasets (LaMP and GOQA), which do not contain personally identifiable information. For the human evaluation study, we recruited crowdworkers via Amazon Mechanical Turk (AMT). All annotators participated voluntarily, were fairly compensated, and no sensitive demographic or personal information was collected.

Our research aims to improve personalization in black-box LLMs by modeling reasoning derived from user history. While this contributes to interpretability and performance, it may also raise concerns regarding privacy, fairness, and potential misuse. To mitigate these risks, our framework is designed to operate on anonymized, structured behavioral data rather than raw personal information, and we explicitly discuss limitations and safeguards in Appendix F.

## 7 REPRODUCIBILITY STATEMENT

We have taken several steps to ensure the reproducibility of our results. Detailed descriptions of the datasets, preprocessing steps, baseline implementations, and evaluation protocols are provided in Appendix D. All prompt templates, factor construction procedures, and algorithmic details are documented in Appendices B, C, and H. To assess robustness, we report results averaged over three runs with standard deviations (Table 5), confirming the stability of our experimental setup. All experiments rely on publicly available datasets (LaMP and GOQA), and the complete prompt configurations and clustering parameters are provided for replication. We also include analyses of hyperparameter choices and ablations Appendix E.

## 8 ACKNOWLEDGMENTS

This work was supported by the IITP grants funded by the Korea government (MSIT) (No.RS-2020-II201361; RS-2024-00457882, AI Research Hub Project).

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

## A  NOTATIONS

Table 4: Notations used throughout the paper.

| Notation | Description |
|---|---|
| $u$ | A user |
| $H_u = \{(q_i, a_i)\}_{i=1}^N$ | History of user $u$ (queries and responses) |
| $q, q'$ | Query and target query |
| $a, a'$ | Response and target response |
| $M$ | Black-box LLM used for feature extraction and generation |
| $G_q$ | Feature set extracted from query $q$ |
| $F^{(m)}$ | A factor obtained by clustering semantically related features |
| $\theta^{(m)}$ | Statistics of factor $F^{(m)}$ (coverage, influence, polarity) |
| $C_u = \{(F^{(m)}, \theta^{(m)})\}$ | Factor set for user $u$ |
| $r_q$ | Personalized reasoning path for query $q$ |
| $S_u$ | Reasoning-augmented memory of user $u$, containing $(q, G_q, r_q, a)$ |
| $\text{Coverage}(F^{(m)})$ | Number of instances in history where factor $F^{(m)}$ appears |
| $\text{Influence}(F^{(m)})$ | Count of cases where $F^{(m)}$ influenced a response |
| $\text{Polarity}(e, F^{(m)})$ | Distribution of polarity labels for features in $F^{(m)}$ |

## B  ALGORITHM DETAILS

Algorithm 1 outlines the overall pipeline of RPM.

---

**Algorithm 1** RPM: Reasoning-Level Personalization for Black-Box LLMs

---

**Input:** user $u$ with history $\mathcal{H}_u = \{(q_i, a_i)\}_{i=1}^N$; target query $q'$; black-box LLM $\mathcal{M}$
**Output:** personalized reasoning $r_{q'}$; personalized response $a'$
**Definitions:** $\mathcal{G}_q$ – feature set from $q$; $\mathcal{C}_u$ – factor set $\{(F^{(m)}, \theta^{(m)})\}$; $\mathcal{S}_u$ – memory of $(q, \mathcal{G}_q, r_q, a)$
**Stage 1 – Personalized Factor Construction**
1:  **for all** $(q_i, a_i) \in \mathcal{H}_u$ **do**
2:      $\mathcal{G}_{q_i} \leftarrow$ FEATUREEXTRACT$_{\mathcal{M}}(q_i)$                        ▷ extract features with $\mathcal{M}$
3:  **end for**
4:  $\{F^{(m)}\}_{m=1}^M \leftarrow$ LLMCLUSTER$_{\mathcal{M}}\left(\bigcup_{(q_i, a_i) \in \mathcal{H}_u} \mathcal{G}_{q_i}\right)$      ▷ semantic grouping of features with $\mathcal{M}$
5:  $\mathcal{C}_u \leftarrow \emptyset$                                          ▷ initialize factor set with statistics
6:  **for all** $F^{(m)} \in \{F^{(m)}\}_{m=1}^M$ **do**
7:      **if** task has discrete classes **then**
8:          $\theta^{(m)} \leftarrow$ COMPUTEPROPENSITY$(F^{(m)})$              ▷ compute propensity via equation 1
9:      **else**
10:          $\text{cov}^{(m)} \leftarrow$ COMPUTECOVERAGE$(F^{(m)})$            ▷ compute coverage via equation 2
11:          $\text{inf}^{(m)} \leftarrow$ COMPUTEINFLUENCE$(F^{(m)})$        ▷ compute influence via equation 3 with $\mathcal{M}$
12:          $\text{pol}^{(m)} \leftarrow$ COMPUTEPOLARITY$(F^{(m)})$          ▷ compute polarity via equation 4 with $\mathcal{M}$
13:          $\theta^{(m)} \leftarrow (\text{cov}^{(m)}, \text{inf}^{(m)}, \text{pol}^{(m)})$              ▷ store three statistics
14:      **end if**
15:      $\mathcal{C}_u \leftarrow \mathcal{C}_u \cup \left(F^{(m)}, \theta^{(m)}\right)$                      ▷ append factor + stats to $\mathcal{C}_u$
16:  **end for**
**Stage 2 – Personalized Reasoning Construction**
17:  **for all** $(q_i, a_i) \in \mathcal{H}_u$ **do**
18:      $r_{q_i} \leftarrow$ GENERATEREASONING$_{\mathcal{M}}(q_i, \mathcal{G}_{q_i}, \mathcal{C}_u, a_i)$      ▷ generate personalized reasoning **with** $\mathcal{M}$
19:      $\mathcal{S}_u \leftarrow \mathcal{S}_u \cup (q_i, \mathcal{G}_{q_i}, r_{q_i}, a_i)$
20:  **end for**
**Stage 3 – Reasoning-Aligned Generation**
21:  $\mathcal{G}_{q'} \leftarrow$ FEATUREEXTRACT$_{\mathcal{M}}(q', \mathcal{C}_u)$                  ▷ extract features from $q'$ with $\mathcal{M}$
22:  $\mathcal{S}_{u,q'}^{\text{ret}} \leftarrow$ RETRIEVETOPK$(\mathcal{S}_u, \mathcal{G}_{q'})$                  ▷ feature-based retrieval via equation 5
23:  $(r_{q'}, a') \leftarrow$ GENERATEREASONING$_{\mathcal{M}}(q', \mathcal{G}_{q'}, \mathcal{C}_u, \mathcal{S}_{u,q'}^{\text{ret}})$  ▷ generate with reasoning-examples with $\mathcal{M}$
24:  **return** $(r_{q'}, a')$

---

## C   LLM-BASED CLUSTERING DETAILS

Among available clustering methods, we adopt the **Propose-Assign-Select (PAS)** framework (Wang et al., 2023), which is an LLM-based goal-driven explainable clustering method. PAS evaluates both a feature's `name` and its `context` with the LLM's powerful semantic understanding capability, enabling the algorithm to group features that affect the user's response in similar ways and to provide a natural-language explanation for every resulting group.

**Feature pool.** For a user history $\mathcal{H}_u = \{(q_i, a_i)\}_{i=1}^N$ we extract feature sets $\{\mathcal{G}_{q_i}\}_{i=1}^N$ (Sec. 3.2) and form their union, denoted $\bigcup_i \mathcal{G}_{q_i}$, as the pool to be clustered.

**Iterative PAS cycle ($\leq R_{\max}$ rounds, $R_{\max} = 3$).** All LLM-based operations in PAS—including proposing candidate factors, assigning features, and handling residuals—are conducted using the same backbone model, `gpt-4o-mini` (Hurst et al., 2024).

At the beginning of each round, we sample a *new random subset* corresponding to 30% of the entire feature pool, restricted to features that have not yet been covered. This subset is used solely in the **Propose** stage to generate a diverse set of candidate factors while keeping the prompt length manageable. The subsequent **Assign** and **Select** stages operate on the entire uncovered feature pool using the full set of candidate factors generated in the current round.

1. **Propose**: The proposer model receives a randomly sampled subset of the uncovered features (covering 30% of the full pool) and the goal prompt, and returns $L$ natural-language candidate factors $\{F_l\}_{l=1}^L$. This subset is used only for factor generation, not for assignment or selection.

2. **Assign**: For each uncovered feature $f$ in the full pool, the assigner model receives the entire set of candidate factors $\{F_l\}_{l=1}^L$ and decides if $f$ matches any of them, and if so, assigns it to the most relevant one. The model is instructed to assign $f$ to at most one semantically most relevant factor, and to skip assignment if no appropriate match exists. Based on the assignment result, we populate the assignment matrix $A \in \{0, 1\}^{|\mathcal{G}| \times L}$, where each row corresponds to a feature and each column to a candidate factor. Specifically, $A(f, F_l) = 1$ indicates that feature $f$ is assigned to candidate factor $F_l$, while $A(f, F_l) = 0$ otherwise.

3. **Select**: To select factors most efficiently without duplication, we iteratively choose those with minimal overlap and maximal coverage, assembling a compact set of highly representative clusters. Specifically, we maintain a set of "remaining" features not yet associated with any chosen factor (initially all features). At each step, we select the factor that covers the largest number of these remaining features, remove those features from the set, and repeat until the remaining set is empty or we have chosen $P_{\max}$ factors. If fewer than 95% of features have been accounted for after selecting $P_{\max}$ factors, we initiate a new round with the remaining unassigned features.

The parameter selection for PAS clustering was guided by a principle of prioritizing generalizability and reproducibility over expensive, dataset-specific hyperparameter tuning. This approach was chosen to demonstrate that the strong performance of RPM stems from the core framework itself, not from fine-grained optimization. Following this principle, the standard, validated parameters from the original cited work were adopted. Specifically, to balance factor representativeness against the typical feature count per user, the number of candidate factors per round was set to $L = 16$ and the maximum factors selected in each iteration was set to $P_{\max} = 8$. This commitment to principled parameter settings ensures the results are both reproducible and provide a robust baseline for future work.

**Handling residual features.** After the iteration, every feature that remains unassigned is re-evaluated by the assigner model with an additional prompt instructing it to assign each residual item to the most semantically suitable existing factor. This step raises the average coverage per user above 99%.

**Factor set.** PAS outputs a set of $M$ factors $\{F^{(m)}\}_{m=1}^M$, where each factor $F^{(m)}$ comprises a factor name and the set of features assigned to it, linking each feature to its corresponding factor. These factors constitute the structure on which we later compute statistical summaries such as propensity, coverage, influence, and polarity for reasoning-level personalization.

Further implementation details are available in the provided code and in the original paper (Wang et al., 2023).

## D    EXPERIMENTAL DETAILS

### D.1    DATASET AND TASK DETAILS

We conduct evaluations on four personalization tasks: classification, regression, generation, and question answering, sourced from the LaMP benchmark (Salemi et al., 2023) and GlobalOpinionQA (GOQA) (Durmus et al., 2023), each of which presents unique challenges for modeling user-specific decision-making patterns.

- **LaMP-2**: Multi-Label Movie Tag Classification. This task involves predicting a single user-assigned tag for a movie based on its description. Each user is associated with a history of previously tagged movies, which serves as their profile. Only the 15 most popular tags from the MovieLens dataset are used as labels. We subsample 50 users from the original time-based validation split. Each user history is partitioned into 36 training and 4 test samples in chronological order. Evaluation is based on accuracy and F1 score.

- **LaMP-3**: Product Rating Prediction. This task involves predicting the 1–5 star rating that a user would assign to a product based on their review. We subsample 100 users with sufficiently long review histories, each split into 90 training and 10 test samples by timestamp. We evaluate model performance using Mean Absolute Error (MAE) and Root Mean Squared Error (RMSE), following the original evaluation protocol of LaMP.

- **LaMP-5**: Scholarly Title Generation. This task aims to generate an academic title for a given paper abstract, reflecting the user's stylistic preferences. Each user represents an author who has written multiple papers, with available abstracts and titles. We sample 100 users, each with 90 training and 10 test samples sorted chronologically. Evaluation is based on ROUGE-1 and ROUGE-L metrics to measure lexical overlap between generated and reference titles.

- **GOQA**: Personalized Question Answering. GOQA is a multiple-choice QA task built on global opinion surveys. Each user corresponds to a demographic group defined by country. The goal is to predict the answer most likely to be selected by a given group for each question. We include only high-confidence samples (where the top answer's selection rate exceeds 0.8), yielding 46 user groups. For each group, we randomly sample 40 responses and split them into 36 training and 4 test samples. Evaluation focuses on accuracy, which reflects alignment with population-level opinions.

### D.2    BASELINES

We compare our proposed method, RPM, against a diverse set of representative baselines for blackbox LLM personalization. All baselines operate under the same API-based constraints and utilize a shared backbone model (`gpt-4o-mini`) for a fair comparison. Unless otherwise noted, the number of retrieved in-context examples is fixed to 3 across methods.

- **Zero-shot**: The target query is directly passed to the language model without any user-specific context. This serves as a non-personalized reference point.

- **In-Context Learning (ICL)**: A few examples from the user's history are inserted into the prompt. These examples are selected randomly, without retrieval or optimization.

- **Retrieval-Augmented Generation (RAG)** (Salemi et al., 2023): Similar to ICL, but the examples are selected using semantic similarity via Contriever, enabling more relevant context injection.

- **Profile-Augmented Generation (PAG)** (Richardson et al., 2023): User histories are summarized into natural-language profile descriptions. We use top-10 retrieved histories for summary generation.

- **HYDRA** (Zhuang et al., 2024): A plug-and-play framework that uses a reranker module to reorder retrieved in-context examples and an adapter module to select the most suitable response from multiple LLM generations.

- **Fermi** (Kim & Yang, 2024): A prompt refinement method that iteratively updates the user prompt using feedback from prior misaligned generations, optimizing input construction over time.

### D.3 IMPLEMENTATION DETAILS

All experiments were conducted on a CPU-only server with an Intel Xeon Gold 6526Y (2.80GHz, 64 cores, 128 threads), using Python 3.10.13.

The black-box language model that serves as the backbone across all experiments is `gpt-4o-mini` (`gpt-4o-mini-2024-07-18`), accessed via the OpenAI API using the LangChain framework[2]. All components of RPM—including feature extraction, factor construction, and reasoning generation—are implemented using dedicated prompt templates designed for each subtask, executed through the API. To ensure deterministic outputs and reproducibility, we fix the decoding temperature to 0.0 for all inference steps across methods. However, minor variations in outputs were occasionally observed, likely attributable to the non-deterministic nature of the API provider's backend services. This configuration is consistently applied to RPM and all baseline methods during evaluation.

Exceptions are made for baseline methods that explicitly rely on sampling-based generation as part of their original design. HYDRA (Zhuang et al., 2024) selects from sampled response candidates using an adapter controller. We follow the original setting and use temperature 1.0 for sampling. Fermi (Kim & Yang, 2024) utilizes prompt optimization via feedback-driven sampling. During its prompt search phase, we also apply a temperature of 1.0 to enable diverse candidate generation. These non-deterministic settings are limited strictly to internal sampling stages defined by the original methods. All final outputs for evaluation are generated with temperature 0.0 to ensure consistency across methods. No maximum token limit was enforced, and no truncation-related issues were observed in any instance.

### D.4 HUMAN EVALUATION

We conduct a human evaluation study on Amazon Mechanical Turk (AMT)[3] to assess the reasoning quality of different methods across all four datasets. For each dataset, we randomly sample 50 examples, and compare outputs from three methods: RPM, Fermi + CoT, and HYDRA + CoT. Each example is evaluated by five independent annotators, who assess the reasoning outputs on the following six criteria:

- **Persuasiveness**: How convincing the reasoning is in supporting the answer.

- **Reasonability**: The logical soundness and coherence of the explanation.

- **Faithfulness**: Whether the reasoning is input-grounded and output-consistent, accurately reflecting the provided input information and remaining logically consistent with the final answer.

- **Interpretability**: How clearly the reasoning shows the connection between input and output.

- **Alignment**: Consistency of the reasoning with the structure of few-shot examples.

- **Overall Quality**: General preference for the best overall explanation.

Annotators select the best and worst explanation per criterion among the three anonymized outputs (labeled A, B, C), enabling stable pairwise comparison.

Additionally, we validate the semantic validity of components constructed by RPM:

- **Feature Validity**: Whether each extracted feature is relevant to the corresponding response.

- **Factor Appropriateness**: Whether each factor appropriately groups features with shared influence.

As shown in Figure 3, RPM achieves strong human preference in interpretability and alignment, and over 90% of features and factors are judged valid. Full annotation guidelines and interface are illustrated in Figure 5 and Figure 6.

Table 5: Overall performance comparison across all benchmarks with standard deviation over 3 runs. For each method, the +CoT variant denotes the use of Chain-of-Thought prompting during inference. RPM (w/o Reasoning) disables reasoning generation for the target query while keeping retrieved reasoning paths in the input context.

| Dataset | | LaMP-2 | | LaMP-3 | | LaMP-5 | | GOQA |
|---|---|---|---|---|---|---|---|---|
| Method | +CoT | Acc. ↑ | F1 ↑ | MAE ↓ | RMSE ↓ | R-1 ↑ | R-L ↑ | Acc. ↑ |
| Zero-shot | | $0.430 \pm 0.008$ | $0.360 \pm 0.007$ | $0.361 \pm 0.008$ | $0.680 \pm 0.011$ | $0.446 \pm 0.001$ | $0.364 \pm 0.001$ | $0.562 \pm 0.014$ |
| | ✓ | $0.411 \pm 0.014$ | $0.337 \pm 0.013$ | $0.323 \pm 0.006$ | $0.630 \pm 0.008$ | $0.434 \pm 0.001$ | $0.376 \pm 0.001$ | $0.557 \pm 0.017$ |
| ICL | | $0.495 \pm 0.018$ | $0.412 \pm 0.017$ | $0.333 \pm 0.003$ | $0.638 \pm 0.003$ | $0.455 \pm 0.002$ | $0.395 \pm 0.002$ | $0.695 \pm 0.005$ |
| | ✓ | $0.471 \pm 0.006$ | $0.374 \pm 0.010$ | $0.317 \pm 0.009$ | $0.625 \pm 0.009$ | $0.460 \pm 0.001$ | $0.405 \pm 0.001$ | $0.681 \pm 0.027$ |
| RAG | | $0.526 \pm 0.010$ | $0.438 \pm 0.012$ | $0.363 \pm 0.003$ | $0.687 \pm 0.004$ | $0.462 \pm 0.001$ | $0.405 \pm 0.001$ | $0.773 \pm 0.008$ |
| | ✓ | $0.493 \pm 0.008$ | $0.415 \pm 0.009$ | $0.366 \pm 0.007$ | $0.690 \pm 0.003$ | $0.469 \pm 0.001$ | $0.412 \pm 0.001$ | $0.800 \pm 0.022$ |
| PAG | | $0.525 \pm 0.013$ | $0.444 \pm 0.026$ | $0.331 \pm 0.006$ | $0.662 \pm 0.007$ | $0.463 \pm 0.002$ | $0.404 \pm 0.003$ | $0.795 \pm 0.006$ |
| | ✓ | $0.513 \pm 0.008$ | $0.431 \pm 0.012$ | $0.339 \pm 0.005$ | $0.672 \pm 0.002$ | $0.464 \pm 0.001$ | $0.405 \pm 0.002$ | $0.820 \pm 0.009$ |
| HYDRA | | $0.526 \pm 0.006$ | $0.437 \pm 0.013$ | $0.324 \pm 0.003$ | $0.656 \pm 0.009$ | $0.463 \pm 0.000$ | $0.406 \pm 0.000$ | $0.800 \pm 0.006$ |
| | ✓ | $0.496 \pm 0.003$ | $0.406 \pm 0.007$ | $0.353 \pm 0.008$ | $0.672 \pm 0.005$ | $0.465 \pm 0.003$ | $0.409 \pm 0.005$ | $0.806 \pm 0.017$ |
| Fermi | | $0.526 \pm 0.012$ | $0.437 \pm 0.008$ | $0.328 \pm 0.034$ | $0.628 \pm 0.030$ | $0.465 \pm 0.007$ | $0.402 \pm 0.006$ | $0.800 \pm 0.008$ |
| | ✓ | $0.476 \pm 0.018$ | $0.377 \pm 0.022$ | $0.312 \pm 0.012$ | $0.635 \pm 0.006$ | $0.453 \pm 0.006$ | $0.395 \pm 0.007$ | $0.659 \pm 0.021$ |
| RPM (w/o Reasoning) | | $0.510 \pm 0.013$ | $0.398 \pm 0.019$ | $0.305 \pm 0.005$ | $0.599 \pm 0.007$ | $0.466 \pm 0.001$ | $0.388 \pm 0.002$ | $0.820 \pm 0.011$ |
| RPM | | $\mathbf{0.561} \pm 0.012$ | $\mathbf{0.463} \pm 0.014$ | $\mathbf{0.259} \pm 0.009$ | $\mathbf{0.548} \pm 0.008$ | $\mathbf{0.492} \pm 0.003$ | $\mathbf{0.416} \pm 0.003$ | $\mathbf{0.852} \pm 0.017$ |

Table 6: Performance comparison across different configurations of `context`-field in the feature. We evaluate how each configuration—`reference`, `evaluation`, `w/o context`, and the original `context` used in our method—affects personalization performance across four benchmarks.

| context-field Configuration | LaMP-2 | | LaMP-3 | | LaMP-5 | | GOQA |
|---|---|---|---|---|---|---|---|
| | Acc. ↑ | F1 ↑ | MAE ↓ | RMSE ↓ | R-1 ↑ | R-L ↑ | Acc. ↑ |
| reference | 0.522 | 0.421 | 0.287 | 0.585 | 0.485 | 0.412 | 0.842 |
| evaluation | 0.530 | 0.429 | 0.286 | 0.574 | 0.483 | 0.408 | 0.826 |
| w/o context | 0.512 | 0.417 | 0.279 | 0.566 | 0.481 | 0.409 | 0.842 |
| context (Ours) | **0.561** | **0.463** | **0.259** | **0.548** | **0.492** | **0.416** | **0.852** |

# E ADDITIONAL ANALYSES

## E.1 STATISTICAL TEST

Table 5 reports the standard deviation of performance metrics across all methods and benchmarks. The results show that all methods exhibit consistently low variance across repeated runs, indicating that the experimental setup is stable and reliable. To minimize stochastic effects during inference, we fixed the decoding temperature to 0.0 for all methods, ensuring deterministic outputs. This design choice allows meaningful comparison between methods and supports the reproducibility of results.

## E.2 FEATURE CONFIGURATION

To construct structured features, we define each feature as a tuple of `name`, `context`, and `factor`, where the `name` identifies the semantic element, the `context` disambiguates and concretizes the feature's intended meaning, and the `factor` links the feature to a higher-level user behavioral pattern with statistics of factors. Among these components, the `context`-field plays a critical role in providing a richer explanation of the feature, which can enhance both clustering quality and personalization performance.

To further examine how different definitions of the feature representation influence RPM's personalization performance, we conduct an extended analysis by modifying the configuration of the `context`-field within each feature triplet. Specifically, we compare the following four configurations:

---

[2] https://www.langchain.com/
[3] https://www.mturk.com/

Table 7: Performance comparison between *feature-level* scoring (matching each feature individually) and our *sample-level* concatenation retrieval across four personalization benchmarks.

| Dataset | LaMP-2 | | LaMP-3 | | LaMP-5 | | GOQA |
|---|---|---|---|---|---|---|---|
| Method | Acc. ↑ | F1 ↑ | MAE↓ | RMSE↓ | R-1 ↑ | R-L ↑ | Acc. ↑ |
| Feature-level | 0.530 | 0.441 | 0.289 | 0.573 | 0.488 | 0.400 | 0.847 |
| Sample-level | **0.561** | **0.463** | **0.259** | **0.548** | **0.492** | **0.416** | **0.852** |

Table 8: Ablation study on the retrieval similarity formulation. **Feature only** uses the raw feature texts, whereas **Factor+Feature** additionally weights candidates by the overlap of their factor sets.

| Dataset | LaMP-2 | | LaMP-3 | | LaMP-5 | | GOQA |
|---|---|---|---|---|---|---|---|
| Method | Acc. ↑ | F1 ↑ | MAE↓ | RMSE↓ | R-1 ↑ | R-L ↑ | Acc. ↑ |
| Factor-Feature | 0.530 | 0.431 | 0.294 | 0.579 | 0.482 | 0.408 | 0.847 |
| Feature only | **0.561** | **0.463** | **0.259** | **0.548** | **0.492** | **0.416** | **0.852** |

- `reference`: the original text span from which the feature is extracted.
- `evaluation`: user sentiment (e.g., positive or negative statements) or explicit evaluative expressions associated with the feature.
- w/o `context`: only the `name` and associated `factor` are retained, omitting the `context`-field.
- `context` (ours): a clarifying phrase that grounds the feature in its surrounding query, providing a disambiguated interpretation of its intended meaning. This is the default configuration used throughout the main RPM pipeline.

We apply each variant throughout the full RPM pipeline—including factor construction, reasoning construction, and reasoning-aligned generation—and report the corresponding personalization performance in Table 6.

While all configurations achieve competitive performance, using the original `context` consistently yields the best results. This suggests that, among various forms of feature configuration, grounding contextual information in the full query provides the most effective disambiguation of feature semantics, thereby yielding improved personalization performance.

### E.3 RETRIEVAL STRATEGY

**Feature-level vs. Sample-level Retrieval.** Our default *sample-level* retrieval strategy embeds each reasoning example with query $q_i$ as a single sequence obtained by concatenating all of its feature texts and then computes a cosine similarity with the target query $q'$, represented as the concatenation of its feature texts. To assess the effect of finer matching granularity, we implement a *feature-level* variant that scores candidates by matching individual features.

Let $\mathcal{G}_{q'} = \{ f'_k \}_{k=1}^{|\mathcal{G}_{q'}|}$ be the feature set of the target query and $\mathcal{G}_{q_i} = \{ f_j \}_{j=1}^{|\mathcal{G}_{q_i}|}$ the feature set of a candidate example with query $q_i$. For every target feature $f'_k$, we compute its cosine similarity to all $f_j$ in $\mathcal{G}_{q_i}$ and keep only the largest value; summing these maxima yields the relevance score of $\mathcal{G}_{q_i}$. We define the feature-level relevance score function $S_{\text{feat}}(q', q_i)$ as:

$$S_{\text{feat}}(q', q_i) = \sum_{k=1}^{|\mathcal{G}_{q'}|} \max_{1 \le j \le |\mathcal{G}_{q_i}|} \cos\big(f'_k, f_j\big),$$

where $\cos(\cdot, \cdot)$ denotes the cosine similarity between two feature embeddings. The score $S_{\text{feat}}(q', q_i)$ measures how well the reasoning example $q_i$ aligns with the feature semantics of the target query $q'$. We evaluate $S_{\text{feat}}(q', q_i)$ for every stored example and retrieve the top-$K$ with the highest scores.

Table 7 confirms that the sample-level concatenation used in RPM consistently outperforms the feature-level variant, which—despite being competitive—often over-weights generic features shared by many samples and fails to leverage how multiple cues jointly characterize the user's behavior

pattern. Also, the feature-level approach incurs a cost of $\mathcal{O}\big(N\,|\mathcal{G}_{q'}|\,\overline{|\mathcal{G}_{q_i}|}\big)$ cosine evaluations ($N$ is the number of stored examples and $\overline{|\mathcal{G}_{q_i}|}$ is the average feature count per example), whereas the sample-level scheme requires only $\mathcal{O}(N)$. Hence the default retrieval is both *more effective* and *far more efficient*.

**Two-Stage Retrieval with Factors.** We also propose a factor-guided two-stage retrieval scheme that exploits the factor identifiers obtained for each feature (Section 3.2). For any query $q$, let $\mathcal{F}_q$ denote the set of factor indices present in its features.

- **Stage 1 (factor filter).** Compute the Jaccard similarity $J(\mathcal{F}_{q'}, \mathcal{F}_{q_i})$ between the factor set of $q'$ and $q_i$ of every stored reasoning example in the user's history. Retain all candidates achieving the maximum Jaccard score; if fewer than $3 \times K$ candidates remain, iteratively add the next-best scored groups until exactly $3 \times K$ candidates are collected (truncating any surplus).

- **Stage 2 (feature scoring).** Apply the same sample-level cosine similarity as in our default method to this reduced pool and select the final top-$K$ reasoning examples.

As reported in Table 8, the factor-guided method attains solid performance—slightly below retrieval solely based on features—while lowering runtime thanks to the inexpensive Jaccard pre-filter. It therefore offers a practical option for large-scale or latency-sensitive deployments.

## E.4 COST ANALYSIS

To directly address concerns about scalability and efficiency, we provide a transparent and comparative analysis of both inference latency and computational cost on the GOQA benchmark. RPM introduces a small increase in per-user inference time compared to lightweight baselines, but this overhead is minimal and brings substantial gains in accuracy and personalization. For the cost analysis, we employ the same model (GPT-4o-mini).

**Inference Latency.** ICL/RAG achieves a per-user inference time of 0.04s, while RPM requires 0.10s, consisting of 0.04s for feature extraction and 0.06s for reasoning-aligned generation. This additional ∼0.06s is required for structured reasoning, and all LLM calls are processed asynchronously to minimize bottlenecks. Importantly, after the initial LLM call per user query, retrieval and ranking proceed with efficient feature-based search, requiring no further LLM calls.

**Computational Overhead.** RPM incurs a one-time preprocessing cost of $0.058 per user, which is substantially lower than prompt-heavy or parameter-tuning approaches such as Fermi ($0.32) and HYDRA ($0.47 + additional parameter training). At inference, the per-instance cost of RPM ($0.0037) is slightly higher than ICL/RAG, but remains significantly lower than advanced baselines while delivering superior accuracy.

**Summary of Trade-off.** Overall, RPM achieves 85.2% accuracy, outperforming all compared methods, while introducing only a minor increase in latency and cost relative to ICL/RAG. At the same time, it remains far more efficient and scalable than high-overhead methods like Fermi and HYDRA, which require repeated prompt optimization or parameter tuning. This demonstrates that building a personalized reasoning-augmented user history once per user provides a practical and effective trade-off between efficiency and accuracy.

Table 9: Token usage and per-user cost on the GOQA benchmark.

| Method | Preprocessing Cost ($) | Inference Cost ($) | Accuracy |
|---|---|---|---|
| Zero-shot | 0 | 0.0002 | 0.562 |
| ICL/RAG | 0 | 0.0007 | 0.695 / 0.773 |
| PAG | 0 | 0.0013 | 0.795 |
| HYDRA | 0.4679 + Param. Training | 0.0028 + Reranker/Adapter | 0.800 |
| Fermi | 0.3204 | 0.0007 | 0.800 |
| RPM (ours) | 0.0581 | 0.0037 | **0.852** |

Table 10: Comparison of performance across different backbone models (`gpt-3.5-turbo`, `gpt-4o`, and `o3-mini`) with and without Chain-of-Thought (+CoT) prompting. Each model is evaluated on LaMP-3 and GOQA datasets. RPM and its transfer variant are reported separately to assess the transferability of constructed personalized reasoning across backbone models.

| Backbone | gpt-3.5-turbo | | | gpt-4o | | | o3-mini | | |
|---|---|---|---|---|---|---|---|---|---|
| Dataset | LaMP-3 | | GOQA | LaMP-3 | | GOQA | LaMP-3 | | GOQA |
| Method | MAE↓ | RMSE↓ | Acc. ↑ | MAE↓ | RMSE↓ | Acc. ↑ | MAE↓ | RMSE↓ | Acc. ↑ |
| Zero-shot | 0.496 | 0.806 | 0.690 | 0.262 | 0.559 | 0.837 | 0.300 | 0.622 | 0.668 |
| +CoT | 0.317 | 0.634 | 0.614 | 0.278 | 0.587 | 0.609 | 0.304 | 0.620 | 0.658 |
| RAG | 0.372 | 0.694 | **0.788** | 0.282 | 0.588 | 0.908 | 0.291 | 0.640 | 0.761 |
| +CoT | 0.444 | 0.776 | 0.777 | 0.334 | 0.656 | 0.848 | 0.315 | 0.655 | 0.777 |
| PAG | 0.351 | 0.660 | **0.788** | 0.299 | 0.635 | 0.859 | 0.361 | 0.681 | 0.810 |
| +CoT | 0.324 | 0.666 | 0.772 | 0.303 | 0.666 | 0.870 | 0.306 | 0.666 | 0.799 |
| RPM | **0.299** | **0.594** | 0.744 | 0.242 | **0.534** | **0.913** | **0.257** | 0.568 | **0.837** |
| RPM w/ transfer | 0.306 | 0.614 | 0.771 | **0.235** | 0.539 | 0.886 | 0.258 | **0.563** | 0.831 |

## E.5 RPM WITH VARIOUS BLACK-BOX LLMS

**RPM on Diverse Black-Box LLMs.** To confirm that our framework is not limited to `gpt-4o-mini`, we conduct the entire pipeline on three additional black-box LLMs: `gpt-3.5-turbo` (weaker model), `gpt-4o` (stronger model), and `o3-mini` (reasoning model) (Achiam et al., 2023; Hurst et al., 2024). Experiments were performed on two complementary benchmarks—**LaMP-3** (review texts with discrete ratings) and **GOQA** (multiple-choice survey questions)—so that both label-defined and label-free tasks were covered under a fixed budget. Table 10 (upper block) shows a consistent trend relative to the default backbone: `gpt-4o` raises scores, `gpt-3.5-turbo` lowers them modestly, and `o3-mini` remains close to `gpt-4o-mini`. Across all backbones, RPM consistently delivers the best overall results.

**Cross-Model Transferability.** We next tested whether the features, factors, and personalized reasoning paths generated once with `gpt-4o-mini` could be reused by other LLMs. Each alternative backbone consumed these artifacts unchanged and performed inference exactly as in Section 3.4. Table 10 (lower block) shows that reusing the personalized reasoning memory built with `gpt-4o-mini` lets each backbone reach, and occasionally exceed, the scores obtained when its own memory is freshly constructed.

## E.6 COMPARISON WITH STRUCTURED REASONING BASELINES

To examine whether generic structured reasoning approaches can achieve comparable personalization to RPM, we implement two additional baselines beyond the CoT variants already reported in Table 1.

**Structured Preference Induction.** This method first summarizes a user's historical preferences into a structured natural language description, then uses this description to guide CoT reasoning during inference. Unlike RPM, the reasoning structure is derived from a static preference summary rather than from per-interaction features and factors.

**ToT-style Prompting.** Inspired by Tree-of-Thought (Yao et al., 2023), this method generates multiple candidate reasoning paths for a given query, evaluates each path against the user's history, and selects the most promising one. This tests whether branching exploration can compensate for the absence of user-specific reasoning structures.

Table 11 reports the results. Both structured reasoning baselines improve over standard CoT in certain tasks but consistently fall short of RPM across all benchmarks. This confirms that the performance gains of RPM stem not from structured prompting per se, but from the automatic discovery of user-specific reasoning structures grounded in features and factors.

Table 11: Comparison with structured reasoning baselines. HYDRA + CoT is included for reference.

| Method | LaMP-2 | | LaMP-3 | | LaMP-5 | | GOQA |
|---|---|---|---|---|---|---|---|
| | Acc.↑ | F1↑ | MAE↓ | RMSE↓ | R-1↑ | R-L↑ | Acc.↑ |
| HYDRA + CoT | 0.496 | 0.406 | 0.353 | 0.672 | 0.465 | 0.409 | 0.806 |
| Structured Pref. Induction | 0.495 | 0.409 | 0.312 | 0.628 | 0.489 | 0.415 | 0.777 |
| ToT-style | 0.496 | 0.406 | 0.353 | 0.672 | 0.465 | 0.409 | 0.806 |
| RPM (Ours) | **0.561** | **0.463** | **0.259** | **0.548** | **0.492** | **0.416** | **0.852** |

### E.7 LENGTH BIAS ANALYSIS

To examine whether the human evaluation results are influenced by verbosity, we conduct an additional length bias analysis on the 200 LaMP-5 examples used in the study. For each comparison among RPM, HYDRA, and Fermi, we identify the system producing the longest explanation and measure how often it was selected as the most reasonable. The longest explanation was chosen in 40.48% of cases, slightly above the random baseline (33.33%), with a small effect size (Cohen's h = 0.158), indicating that the deviation is negligible.

We further compare explanation lengths across systems, as shown in Table 12. The mean and standard deviation of explanation lengths are comparable across RPM, HYDRA, and Fermi, suggesting that performance differences cannot be attributed to verbosity alone. Overall, these results suggest that RPM 's improvements are not driven by longer explanations but by stronger grounding and alignment in personalized reasoning.

Table 12: Explanation Length Statistics (LaMP-5, 200 samples)

| System | Mean Length | Std. Dev. |
|---|---|---|
| RPM | 133.00 | 14.15 |
| HYDRA | 128.77 | 16.45 |
| Fermi | 121.01 | 21.81 |

### E.8 HALLUCINATION ANALYSIS IN PERSONALIZED REASONING PATHS

Hallucination is a fundamental limitation in any LLM-based system. While the faithfulness evaluation in Section 4.4 provides indirect evidence that RPM 's reasoning paths are well-grounded in user history, we conduct an additional focused human study to directly measure hallucination rates in personalized reasoning paths. We sample 200 reasoning paths, with 50 randomly selected from each dataset (GOQA, LaMP-2, LaMP-3, and LaMP-5). Two annotators independently check whether any feature or factor mentioned in each reasoning path is absent from the input context provided to the model, including retrieved past reasoning samples, extracted features, and constructed factors. If a reasoning path references elements not present in the input context, it is marked as containing hallucination. The results are summarized in Table 13.

With only 3.5–4.75% hallucination rates and 92.5% inter-annotator agreement, these results provide empirical evidence that generating reasoning paths with explicit reference to structured representations, including extracted features and constructed factors, helps reduce hallucinations. Unsupported content is therefore infrequent and can be systematically identified through inspection of the structured feature–factor representation.

## F LIMITATIONS

**Modeling User-Specific Decision Making Patterns.** Since a user's actual internal thought process is not directly observable, the proposed framework aims to construct a pragmatic approximation of it. This effort involves leveraging accessible data—such as behavioral patterns, contextual information, and responses from user history—to build an explicit and interpretable reasoning model. The effectiveness of this approach is demonstrated in two ways: first, the resulting representations

Table 13: Hallucination Rates in Personalized Reasoning Paths

| Dataset | Annotator 1 (%) | Annotator 2 (%) | Agreement (%) |
|---------|-----------------|-----------------|---------------|
| GOQA | 8.0 | 6.0 | 86.0 |
| LaMP-2 | 6.0 | 9.0 | 88.0 |
| LaMP-3 | 0.0 | 4.0 | 96.0 |
| LaMP-5 | 0.0 | 0.0 | 100.0 |
| Overall | 3.5 | 4.75 | 92.5 |

consistently yield significant personalization performance gains against baselines. Second, a dedicated human evaluation study validated the plausibility and faithfulness of the generated reasoning.

**Computation Cost.** The full pipeline of RPM calls the LLM multiple times, so a non-trivial computational cost is unavoidable. Even when the entire workflow runs on `gpt-4o-mini`, one of the most affordable commercial models, the resulting personalization performance remains comparable to our default backbone, and the same personalized memory can be re-used by the stronger `gpt-4o` without reconstruction, demonstrating that strong cross-model transferability (Table 10) offsets the one-time construction cost. A detailed analysis (Appendix E.4) further confirms that RPM incurs far lower API cost than the strongest prompt-optimization baseline while delivering higher quality of personalization.

**Out-of-Domain Generalization.** RPM captures user-level behavioral tendencies through its factor representation, and these tendencies can transfer to adjacent tasks that share similar decision-making contexts. However, when the target domain is entirely unrelated to any previously observed interaction, meaningful personalization becomes difficult because no relevant behavioral signal exists in the user history. This limitation is not specific to RPM; any black-box personalization method that relies on historical user behavior faces the same constraint when the new domain lacks structural or behavioral overlap with past interactions.

**Factor Set Coverage.** RPM maintains a fixed set of factors constructed from the user's observed history and handles new features through soft-assignment to the closest existing factor. While this mechanism provides reasonable coverage in practice, it does not explicitly detect when accumulated novel features diverge sufficiently from existing factors to warrant reconstruction. Developing principled criteria for triggering factor set updates remains an open direction for future work.

**Data Privacy.** All user inputs and personalized artifacts stay entirely within the provider-hosted API, avoiding any third-party sharing and thereby preserving confidentiality throughout processing. Yet the retrieved history or even the query text itself may still contain sensitive details, leaving a non-zero risk of inadvertent disclosure.

# G   LARGE LANGUAGE MODEL USAGE

In this research, we employed a large language model exclusively for limited editorial assistance, specifically to address minor grammatical errors in our writing. The LLM did not contribute to any aspect of the paper's conceptual framework, structural organization, or intellectual content development.

You are invited to participate in a short survey designed to compare the outputs of **three different reasoning systems**.

In this study, you will be presented with three explanations (labeled Reasoning A, B, and C), each generated by a different system based on the same input.
Your task is to **evaluate and compare these outputs from multiple perspectives**.
For each criterion, you will select the **best and worst** performing explanations.
You will also be asked to assess whether the reasoning accurately uses the given features and assigns them to appropriate factors.
There are no right or wrong answers. Please rely on your own judgment and understanding when making your selections.
Thank you for your participation!

- **Features** are key details extracted from each question that likely affect the user's answer (e.g., product taste, health benefits).

- **Factors** are broader categories that group similar features together (e.g., practicality, insight).

**Question Input**
${question}
**Reference Answer**
${answer}
**Extracted Features**
${features}
**Feature-to-Factor Assignments**
${factors}

| *Reasoning A* | *Reasoning B* | *Reasoning C* |
|---|---|---|
| ${prime} | ${hydra} | ${fermi} |

Figure 5: The instruction and annotation guidelines provided within the human evaluation interface.

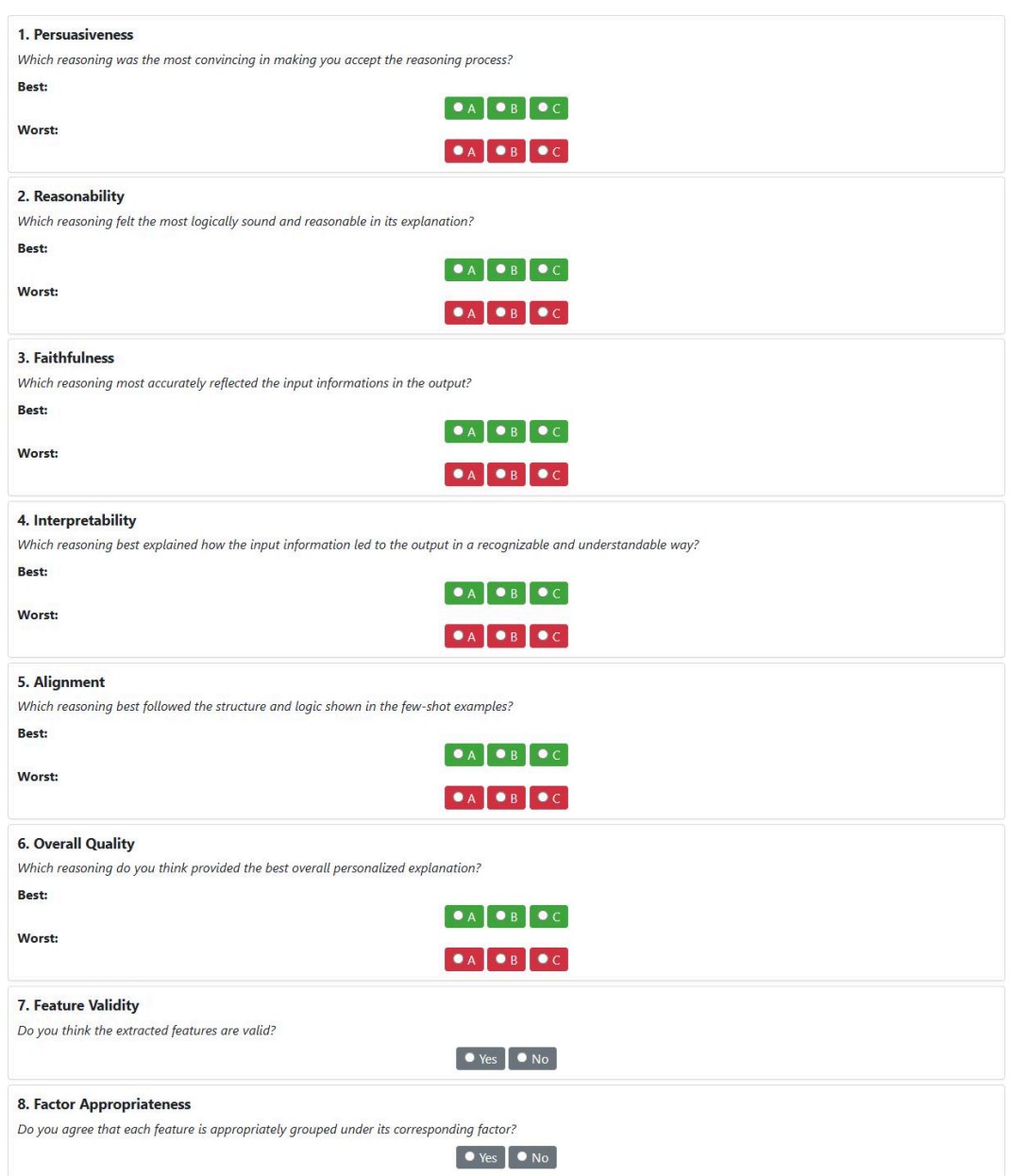

Figure 6: The evaluation form used to compare reasoning outputs across multiple criteria.

# H    PROMPTS

To show the precise instruction prompts we provided to the model on the LaMP-5 benchmark, we present them as follows. Table 14 presents the core prompt template designed for generalizability. Table 15 shows the prompt used to extract all potentially response-influential features from the raw query. Table 16 shows the prompt for proposing factor candidates by clustering the extracted features. Table 17 shows the prompt for assigning each feature to its most semantically relevant factor. Table 18 shows the prompt for evaluating whether each factor influenced the actual response and determining its polarity. Table 19 shows the prompt for building personalized reasoning paths based on features and factors. Table 20 shows the prompt for reasoning-aligned generation with given reasoning examples and target query.

Table 14: Example Prompt Template.

| |
|---|
| Exemplars: {reasoning_examples} 
 You are an expert in personalized [TASK]. 
 Your task is to predict [TASK_OUTPUT] based on their previous preferences and [TASK_INPUT]. |
| TASK_DESCRIPTION: A brief, one-sentence description of the task's objective. [TASK_DESCRIPTION] 
 Analyze the person's preference factors and statistics: 
 - Identify which factors strongly influence this person's [TASK_OUTPUT] preferences 
 - Note the typical patterns associated with each factor 
 - Consider the person's historical [TASK_OUTPUT] preferences as your baseline 
 - Compare this abstract with the similar examples 
 - Look for patterns in how specific features influenced [TASK] in the past 
 - Analyze which reasoning path worked well in previous successful predictions 
 - Consider how the previous reasoning might apply to this [TASK_INPUT] |
| Develop your title prediction by: 
 - Starting with an understanding of the person's historical [TASK_OUTPUT] style as a baseline 
 - Applying successful personalized reasoning from exemplars 
 - Ensuring the [TASK_OUTPUT] accurately reflects the content of the abstract while matching the person's style |
| Return as JSON: { "reasoning": "", "predicted_title": "" } 
 User Preference Factors and Statistics: {factors} 
 [TASK_INPUT]: {task input} 
 [TASK_INPUT] Features: {features} |

Table 15: Feature extraction prompt for LaMP-5.

**Prompt: Feature Extraction**

Extract all relevant features from the paper abstract that could influence its scholarly title generation. Include both explicit features directly mentioned in the text and implicit features that likely influenced the scholarly title generation.
Title and abstract are from dataset that includes information about scientific papers.

For each feature:
1. **Feature Name**: Specific term or concept from the abstract
2. **Context**: The context in which this feature appears

Return as JSON:

```
{{
  "features": [
    {{
      "feature_name": "",
      "context": ""
    }},
    ... (other features)
  ]
}}
```
Title is free-form text strings representing academic paper titles.
Following the given instructions, extract features for the following abstract of a paper:

**Abstract**: {abstract}

Table 16: Factor proposal prompt for LaMP-5.

**Prompt: Factor Proposal**

You are an expert in feature categorization.

Your task is to identify {num_factors} meaningful factors that best categorize the following user features.

Specifically, the goal of the task is to propose the {num_factors} meaningful factors from the features, which are extracted from the paper abstract, that influences the scholarly paper title.

**GUIDELINES**:
- Focus on creating distinct, non-overlapping factors
- Each factor should be 1 word, clear and descriptive
- Factors should be meaningful categories that group similar features
- Aim for general factors that apply across different contexts
- Prioritize FUNDAMENTAL factors that cannot be further reduced
- Ensure factors are ORTHOGONAL to each other (minimal conceptual overlap)
- Create factors with high EXPLANATORY POWER across multiple domains
- Avoid generic labels like "General Factor" or "Dimension X"
- Create factors that would help understand user preferences

**HIERARCHICAL SELECTION PROCESS**:
1. Identify primary evaluation aspects in the features
2. Group features that share fundamental judgment criteria
3. Name each group with the most essential concept that unifies them
4. Test each factor for distinctness from other factors
5. Verify each factor applies across multiple domains

RESPONSE FORMAT:
Return a JSON object with the following structure:

```
{{
  "factors": [
    "factor1",
    "factor2",
    ...
  ]
}}
```
Provide EXACTLY {num_factors} factors that best organize these features.

Following the given instructions, analyze these features from paper abstract and propose {num_factors} meaningful categorization factors that influence the scholarly paper title:

**Feature Examples**: {feature_examples}
**Previous Factors**: {prev_factors}

Table 17: Factor assignment prompt for LaMP-5.

---

**Prompt: Factor Assignment**

---

Your task is to assign a feature to appropriate factors category.

Note: These features were previously extracted from scholary paper abstract using a systematic process to identify both explicit and implicit aspects that influence its paper title generation.

The available factors were generated through hierarchical clustering of these features to create meaningful, orthogonal categories that capture fundamental aspects across scholary title generation.

Each feature should generally be mappable to one of these factors as they were derived from the same underlying data, so try to find the best match even if it's not immediately obvious.

**SYSTEMATIC ASSIGNMENT PROCESS**:
1. Identify the primary evaluative aspect in the feature
2. Extract the MAIN underlying judgment criterion
3. Match this criterion to the factor that BEST represents it
4. Verify this factor captures the CORE ESSENCE of the evaluation

**GUIDELINES**:
- Assign the feature to the factor that best represents it
- Choose the factor that most closely matches the feature's main characteristic

Return your assignment as a JSON object with this structure:

```
{{
  "assignments": ""
  // numbers corresponding to the available factors
}}
```
Following the systematic assignment process, analyze the given feature and assign it to appropriate factor category:

**Feature**: {feature}
**Available Factors**: {proposed_factors}

---

Table 18: Statistical meaning assignment prompt for LaMP-5.

**Prompt: Assigning Statistical Meaning to Factor**

Analyze which features below directly influenced the scholarly title generation.

For each feature, determine whether it influenced the title, and if so, evaluate whether the influence was positive (supporting), negative (opposing), or neutral.

Return as JSON:

```
{{
  "reasoning": "your detailed reasoning here",
  "influences": [
    {{
      "feature_index": 0,
      // index of the feature (0 for first feature in the list, 1 for second,
      etc.)
      "influenced": true,
      // boolean: true if this feature influenced the options and answers,
      false if not
      "evaluation": "pos"
      // If influenced is true, include "pos" for positive influence,
      "neg" for negative influence, "neu" for neutral influence
    }},
    // Include an object for each feature in the input
  ]
}}
```
**Title**: {title}
**Features**: {features}

Table 19: Personalized reasoning construction prompt for LaMP-5.

**Prompt: Personalized Reasoning Construction**

You are an expert in scholarly paper analysis.

Your task is to generate a logical personalized reasoning path that explains how a researcher would arrive at a specific title for a scholarly paper.

**Paper Abstract**: A comprehensive narrative of the paper's research question, methodology, findings, and implications.
**Features**: Specific implicit/explicit elements in a paper abstract that can influence judgment and decision-making in the personalized scholarly title generation process.
**Factors**: General elements that provide statistical measurements about researcher preferences and behaviors, influencing their judgment and decision-making in the personalized scholarly title generation process.
**Title**: The specific scholarly paper title that the researcher would select based on their unique interpretation and personal emphasis of the paper's content.

Create a logical, step-by-step reasoning process that is personalized to the researcher. Your reasoning should:
1. Use statistical factors as the foundation for your reasoning process
2. Build on researcher preferences and patterns revealed in the factors
3. Use features to develop more nuanced, paper abstract-specific step-by-step reasoning
4. Create a logical path showing how the researcher's focus on certain elements leads to their title generation
5. Base reasoning ONLY on the given information (paper abstract, features, factors)
6. Ensure your reasoning would be valid even without knowing the actual title

Use the actual title as a reference point to determine which aspects(features, factors) of the paper abstract the researcher might focus on, but DO NOT mention or use the actual title directly in your reasoning.

Your reasoning should naturally lead to the title with given paper abstract, features and factors without explicitly referencing it(actual title).

Format your response as a JSON object with the following structure:

```
{{
  "reasoning": ""
}}
```

Following the given instructions, analyze these features, factors and generate a personalized reasoning based on them:

**Paper Abstract**: {abstract}
**Features**: {features}
**Factors**: {factors}
**The actual title for this paper**: {title}

Table 20: Reasoning aligned generation prompt for LaMP-5.

**Prompt: Reasoning Aligned Generation**

Exemplars: {reasoning_examples}

You are an expert in personalized academic paper title generation.

Your task is to predict how a person would title a research paper based on their previous preferences and academic writing style.

Generate a personalized title for the following research paper abstract that matches the person's preferences and title patterns.

1. Analyze the person's preference factors and statistics:
- Identify which factors strongly influence this person's title preferences
- Note the typical patterns associated with each factor
- Consider the person's historical title preferences as your baseline

2. Compare this abstract with the similar examples
- Look for patterns in how specific features influenced titles in the past
- Analyze which reasoning path worked well in previous successful predictions
- Consider how the previous reasoning might apply to this abstract

3. Develop your title prediction by:
- Starting with an understanding of the person's historical title style as a baseline
- Applying successful personalized reasoning from exemplars
- Ensuring the title accurately reflects the content of the abstract while matching the person's style

Return as JSON:

```
{{
    "reasoning": "",
    "predicted_title": ""
}}
```
**User Preference Factors and Statistics**: {factors}
**Abstract**: {abstract}
**Abstract Features**: {features}

# I    CASE STUDIES

To illustrate how our model performs personalized reasoning in practice, we present a series of case studies based on the LaMP-5 benchmark. In this task, the input query is a paper abstract, and the expected output response is a scholarly title. We show how the model extracts salient features from each abstract, maps them to structured factors, and ultimately generates a personalized reasoning path that supports the predicted title.

The following tables provide a step-by-step view of this process. Table 21 presents the features extracted from the query (abstract) in Example 1. Table 22 summarizes the user-level factors structured from previously extracted features in the profile. Table 23 shows the personalized reasoning generated for Example 1, based on the query and gold response (i.e., the original title). Table 24 displays the model-generated reasoning and final response (predicted title) for the given target query. Table 25 compares the reasoning and responses produced by RPM, HYDRA, and Fermi for the same target query. Together, these tables illustrate how the model builds and applies user-specific reasoning paths from raw input to final output, enabling both interpretability and personalization.

Table 21: Extracted features from the input query (abstract). Each feature is represented in the format {*feature name : context*}.

| Query (Abstract) |
| --- |
| Mobile crowd-sensing applications produce useful knowledge of the surrounding environment, which makes our life more predictable. However, these applications often require people to contribute, consciously or unconsciously, location-related data for analysis, and this gravely encroaches users' location privacy. Aggregate processing is a feasible way for preserving user privacy to some extent, and based on the mode, some privacy-preserving schemes have been proposed. However, existing schemes still cannot guarantee users' location privacy in the scenarios with low density participants. Meanwhile, user accountability also needs to be considered comprehensively to protect the system from malicious users. In this paper, we propose a participant-density-aware privacy-preserving aggregate statistics scheme for mobile crowd-sensing applications. In our scheme, we make use of multi-pseudonym mechanism to overcome the vulnerability due to low participant density. To further handle sybil attacks, based on the Paillier cryptosystem and non-interactive zero-knowledge verification, we advance and improve our solution framework, which also covers the problem of user accountability. Finally, the theoretical analysis indicates that our scheme achieves the desired properties, and the performance experiments demonstrate that our scheme can achieve a balance among accuracy, privacy-protection and computational overhead. |

| Extracted Features |
| --- |
| **Mobile crowd-sensing applications** : Introduces the main subject of the paper, indicating the area of focus. |
| **Location privacy** : Identifies a critical issue that the proposed solution aims to address. |
| **Aggregate processing** : Introduces a technique relevant to the privacy concerns in mobile crowd-sensing. |
| **Privacy-preserving schemes** : Sets the stage for discussing the limitations of current solutions. |
| **Participant-density-aware** : Defines the unique aspect of the proposed solution that differentiates it from existing schemes. |
| **Multi-pseudonym mechanism** : Details a technical approach to enhance privacy in low-density scenarios. |
| **Sybil attacks** : Highlights a security concern that is relevant to user accountability. |
| **Paillier cryptosystem** : Indicates the technical foundation of the proposed scheme. |
| **User accountability** : Emphasizes the need to protect the system from malicious users. |
| **Theoretical analysis and performance experiments** : Describes the evaluation of the proposed solution's effectiveness. |

Table 22: User-level factors with statistics, aggregated from features in the user profile.

**Features from Example 1**

**Mobile crowd-sensing applications** : Introduces the main subject of the paper, indicating the area of focus.
**Location privacy** : Identifies a critical issue that the proposed solution aims to address.
**Aggregate processing** : Introduces a technique relevant to the privacy concerns in mobile crowd-sensing.
**Privacy-preserving schemes** : Sets the stage for discussing the limitations of current solutions.
**Participant-density-aware** : Defines the unique aspect of the proposed solution that differentiates it from existing schemes.
**Multi-pseudonym mechanism** : Details a technical approach to enhance privacy in low-density scenarios.
**Sybil attacks** : Highlights a security concern that is relevant to user accountability.
**Paillier cryptosystem** : Indicates the technical foundation of the proposed scheme.
**User accountability** : Emphasizes the need to protect the system from malicious users.
**Theoretical analysis and performance experiments** : Describes the evaluation of the proposed solution's effectiveness.

**Features from Example 2**

**3D model retrieval** : The context is the growing popularity of 3D models and the necessity for improved retrieval methods.
**Sketch-based approach** : This feature highlights the innovative aspect of the retrieval method being based on sketches.
**Combined line rendering technique** : This feature indicates the technical approach taken in the retrieval process.
**Descriptor based on orientation of feature lines** : This feature emphasizes the analytical aspect of the method, which is crucial for matching.
**Offline and online processing stages** : This feature outlines the structure of the proposed method, indicating its complexity.
**Similarity measurement** : This feature is critical for understanding how the retrieval process operates.
**Preference viewpoints selection** : This feature indicates a refinement step in the retrieval process.
**Robustness against variations** : This feature highlights the effectiveness and reliability of the proposed method.
**Comparison with DTF-SC** : This feature indicates the competitive nature of the research and its validation.
**Higher precision** : This feature underscores the success of the proposed method in achieving better retrieval accuracy.

**Features from Example 3**

...

**User-Specific Factors with Statistics**

| Factor | Count | Directly Influenced (%) | Positive (%) | Neutral (%) | Negative (%) |
|---|---|---|---|---|---|
| Methodology | 86 | 83/86 (96.5%) | 176/183 (96.2%) | 7/183 (3.8%) | 0 (0.0%) |
| Evaluation | 73 | 56/73 (76.7%) | 132/137 (96.4%) | 5/137 (3.6%) | 0 (0.0%) |
| Challenges | 58 | 48/58 (82.8%) | 68/76 (89.5%) | 4/76 (5.3%) | 4/76 (5.3%) |
| Algorithms | 35 | 30/35 (85.7%) | 44/47 (93.6%) | 3/47 (6.4%) | 0 (0.0%) |
| Performance | 65 | 42/65 (64.6%) | 82/88 (93.2%) | 5/88 (5.7%) | 1/88 (1.1%) |

Table 23: Personalized reasoning generated based on the query (abstract) and the gold response (title).

| Component | Content |
|---|---|
| **Query (Abstract)** | Mobile crowd-sensing applications produce useful knowledge of the surrounding environment, which makes our life more predictable. However, these applications often require people to contribute, consciously or unconsciously, location-related data for analysis, and this gravely encroaches users' location privacy. Aggregate processing is a feasible way for preserving user privacy to some extent, and based on the mode, some privacy-preserving schemes have been proposed. However, existing schemes still cannot guarantee users' location privacy in the scenarios with low density participants. Meanwhile, user accountability also needs to be considered comprehensively to protect the system from malicious users. In this paper, we propose a participant-density-aware privacy-preserving aggregate statistics scheme for mobile crowd-sensing applications. In our scheme, we make use of multi-pseudonym mechanism to overcome the vulnerability due to low participant density. To further handle sybil attacks, based on the Paillier cryptosystem and non-interactive zero-knowledge verification, we advance and improve our solution framework, which also covers the problem of user accountability. Finally, the theoretical analysis indicates that our scheme achieves the desired properties, and the performance experiments demonstrate that our scheme can achieve a balance among accuracy, privacy-protection and computational overhead. |
| **Gold Response (Actual Title)** | Participant-Density-Aware Privacy-Preserving Aggregate Statistics for Mobile Crowd-Sensing |
| **Personalized Reasoning** | To generate a title for the scholarly paper, the researcher begins by analyzing the abstract, which outlines the context of mobile crowd-sensing applications and the critical issue of location privacy. Given the statistical factors indicating a strong preference for evaluation (76.7% directly influenced), the researcher recognizes the importance of addressing user privacy concerns as a central theme. The mention of 'aggregate processing' as a methodology for preserving privacy further emphasizes the need for a solution that balances privacy and functionality. The researcher notes the unique aspect of the proposed scheme being 'participant-density-aware,' which directly addresses the challenges posed by low participant density in crowd-sensing scenarios. This focus aligns with the researcher's pattern of emphasizing methodologies that tackle specific challenges, as indicated by the high influence of the 'Challenges' factor (82.8% directly influenced). Additionally, the introduction of the 'multi-pseudonym mechanism' and the handling of 'sybil attacks' highlights the technical depth of the proposed solution, which the researcher finds essential to include in the title. The researcher also considers the implications of user accountability, which is a significant concern in the context of privacy-preserving schemes. By synthesizing these elements—mobile crowd-sensing, privacy preservation, participant density, and the specific methodologies employed—the researcher crafts a title that encapsulates the essence of the paper while reflecting their focus on the innovative aspects of the proposed solution. |

Table 24: Reasoning and predicted response (title) during inference for the target query (abstract). For simplicity, the extracted features are shown in the format *feature name (assigned factor)*.

| Component | Content |
|---|---|
| **Query Example 1 (Abstract 1)** | Mobile crowd-sensing applications produce useful knowledge of the surrounding environment, which makes our life more predictable. However, these applications often require people to contribute, consciously or unconsciously, location-related data for analysis, and this gravely encroaches users' location privacy. ... |
| **Features from Example 1** | **Mobile crowd-sensing applications**: Introduces the main subject of the paper. **Location privacy**: Identifies the core issue addressed. **Aggregate processing**: Presents a technique for privacy preservation. **Privacy-preserving schemes**: Frames prior limitations. **Participant-density-aware**: Highlights novel contribution. **Multi-pseudonym mechanism**: Describes technical solution. **Sybil attacks**: Points out a relevant security threat. **Paillier cryptosystem**: Technical foundation of the scheme. **User accountability**: Describes protection against malicious users. **Theoretical analysis and performance experiments**: Evaluation of effectiveness. |
| **Personalized Reasoning for Example 1** | To generate a title for the scholarly paper, the researcher begins by analyzing the abstract, which outlines the context of mobile crowd-sensing applications and the critical issue of location privacy. ... |
| ... | |
| **Target Query (Abstract)** | Person re-identification is an extremely challenging problem as person's appearance often undergoes dramatic changes due to the large variations of viewpoints, illuminations, poses, image resolutions, and cluttered backgrounds. How to extract discriminative features is one of the most critical ways to address these challenges. In this paper, we mainly focus on learning high-level features and combine the low-level, mid-level, and high-level features together to re-identify a person across different cameras. Firstly, we design a Siamese inception architecture network to automatically learn effective semantic features for person re-identification in different camera views. Furthermore, we combine multi-level features in null space with the null Foley–Sammon transform metric learning approach. In this null space, images of the same person are projected to a single point, minimizing intra-class scatter and maximizing inter-class separation. Comprehensive evaluations demonstrate improved performance on four benchmark datasets: Market-1501, CUHK03, PRID2011, and VIPeR. |
| **Extracted Features** | *Person re-identification* (Challenges), *Multiple camera views* (Challenges), *Discriminative features* (Methodology), *Deep learning approach* (Methodology), *Siamese CNN* (Methodology), *Null Foley–Sammon Transform* (Methodology), *Recognition performance* (Performance, Methodology), *Market-1501 dataset* (Evaluation), *State-of-the-art methods* (Performance, Methodology) |
| **Personalized Reasoning** | To generate a title for the scholarly paper, I begin by analyzing the abstract, which emphasizes the challenges of person re-identification due to variations in appearance. The researcher has a strong preference for methodologies, as indicated by the high influence rate (96.5%) in the Methodology category. This suggests that the researcher values innovative approaches and techniques in their work. The abstract highlights the use of a Siamese inception architecture network, which is a unique and innovative method for learning features, aligning with the researcher's preference for novel methodologies. The combination of multi-level features and the null Foley–Sammon transform metric learning approach further emphasizes the technical depth of the proposed solution, which the researcher likely wants to reflect in the title. The mention of comprehensive evaluations and better performance on benchmark datasets indicates a focus on performance metrics, which is also a significant aspect of the findings. Therefore, I synthesize these elements—Siamese inception architecture, multi-level feature learning, and performance improvement—into a concise title that reflects the core contributions of the paper while matching the researcher's preferences. |
| **Pred. Response (Pred. Title)** | Siamese Inception Architecture for Multi-Level Feature Learning in Person Re-Identification |
| **Gold Response (Actual Title)** | A Siamese inception architecture network for person re-identification. |

Table 25: Comparison of reasoning and responses (titles) from RPM, HYDRA, and Fermi.

| Component | Content |
|---|---|
| **HYDRA + CoT** | |
| **Reasoning** | The abstract discusses a method for person re-identification that focuses on learning and combining features at multiple levels (low, mid, and high) using a Siamese inception architecture network and a null space approach. The title should reflect the main focus of the research, which is person re-identification, and highlight the specific techniques used, such as the Siamese network and the null space metric learning. The title should also indicate that the approach is based on deep learning, as this is a significant aspect of the methodology. Therefore, a suitable title would be 'A Deep Learning-Based Approach for Person Re-Identification Using Multi-Level Feature Combination in Null Space'. This title captures the essence of the research while being concise and informative. |
| **Response (Title)** | A Deep Learning-Based Approach for Person Re-Identification Using Multi-Level Feature Combination in Null Space |
| **Fermi + CoT** | |
| **Reasoning** | The abstract discusses a method for person re-identification that involves learning and combining features at different levels (low, mid, high) using a Siamese inception architecture and a specific metric learning approach. The title should reflect the core focus of the paper, which is on the advanced techniques used for person re-identification across different camera views. The title 'Optimizing Air Quality Prediction through Advanced Surveillance Camera Techniques and Multimodal Image Analysis' does not align with the content of the abstract, as it mentions air quality prediction and multimodal image analysis, which are not the main topics of the paper. A more appropriate title would focus on person re-identification, feature learning, and the specific methodologies employed. Therefore, a suitable title could be 'Advanced Feature Learning for Person Re-Identification Across Multiple Camera Views'. |
| **Response (Title)** | Advanced Feature Learning for Person Re-Identification Across Multiple Camera Views |
| **RPM (Ours)** | |
| **Reasoning** | To generate a title for the scholarly paper, I begin by analyzing the abstract, which emphasizes the challenges of person re-identification due to variations in appearance. The researcher has a strong preference for methodologies, as indicated by the high influence rate (96.5%) in the Methodology category. This suggests that the researcher values innovative approaches and techniques in their work. The abstract highlights the use of a Siamese inception architecture network, which is a unique and innovative method for learning features, aligning with the researcher's preference for novel methodologies. The combination of multi-level features and the null Foley–Sammon transform metric learning approach further emphasizes the technical depth of the proposed solution, which the researcher likely wants to reflect in the title. The mention of comprehensive evaluations and better performance on benchmark datasets indicates a focus on performance metrics, which is also a significant aspect of the findings. Therefore, I synthesize these elements—Siamese inception architecture, multi-level feature learning, and performance improvement—into a concise title that reflects the core contributions of the paper while matching the researcher's preferences. |
| **Response (Title)** | Siamese Inception Architecture for Multi-Level Feature Learning in Person Re-Identification |
| **Gold Response (Actual Title)** | A Siamese inception architecture network for person re-identification. |

