# OpenReview forum: "RPM: Reasoning-Level Personalization for Black-Box Large Language Models"
_ICLR.cc/2026/Conference — ICLR 2026 Poster_

### Official Review · Reviewer_xTTK · 2025-10-26

**Soundness:** 3
**Presentation:** 2
**Contribution:** 3
**Rating:** 6
**Confidence:** 3

**Summary:**

This paper addresses the limitations of the black-box LLM Response-Level Personalization, which leads to superficial pattern learning and a lack of interpretability. Accordingly, it proposes Reasoning-Level Personalization, which extracts features from user histories, clusters them into quantifiable factors, constructs a structured user model, and generates explicit reasoning paths for each historical interaction. Experimental results demonstrate that the proposed RPM consistently outperforms existing response-level methods in terms of accuracy and exhibits improved interpretability.

**Strengths:**

1. The introduction of "reasoning-level personalization" is reasonable and meaningful.
2. An interpretable personalization framework is proposed.
3. Comparisons with baseline methods and ablation experiments demonstrate the effectiveness of the proposed strategy and the rationality of each module.

**Weaknesses:**

1. The definition of structured features is confusing and unclear until the appendix. However, the main text provides no further guidance.
2. Explanations should be faithful to the model's decisions and cannot be verified through human evaluation.
3. Although the inference cost in Table 9 appears negligible (i.e., 0.0037), for a large service provider, this represents a cost increase of several times, or even dozens of times, compared to the baseline method. This requires further discussion.

**Questions:**

1. Does the preprocessing process refer to the personalized factor construction and personalized inference building process?

---

> ### Author Response · Authors · 2025-11-20
> **Response #1**
>
> We thank the reviewer for the thoughtful feedback. We address each concern below with supporting evidence and additional analyses.
>
> ### **Issue1: The definition of structured features is confusing and unclear until the appendix.**
>
> We appreciate the reviewer’s feedback. Section 3.2 already introduces the components used in feature extraction and explains how they are mapped to factor statistics and the user model, but we agree that the definition of structured features is not presented explicitly enough in the main text. **To improve clarity, we will revise Section 3.2 to provide a concise and self-contained definition of structured features, including how each feature is represented and how these representations contribute to factor construction.** This revision will allow readers to understand the concept without needing to consult the appendix.
>
> ### **Issue 2: Explanations should be faithful to the model’s decisions and cannot be verified through human evaluation.**
>
> We appreciate the reviewer’s concern. In a black-box LLM setting, the model’s internal computations and the internal factors influencing its decisions are not directly observable. Because of this limitation, **the most appropriate form of faithfulness is one that is grounded in observable evidence and consistent with the model’s output.**
>
> RPM is designed to satisfy this type of faithfulness. **For each query, the model jointly generates the reasoning and the final answer using the same input evidence:** the query, extracted features, user-level factors, and retrieved past interactions. Since these components come from the user’s history, the reasoning is grounded in available inputs. Joint generation also ensures that the reasoning remains consistent with the model’s output, **generating a coherent pathway from input to decision and achieving input-grounded and output-consistent faithfulness.**
>
> Human evaluation does not attempt to infer hidden internal states. Instead, raters assess whether the explanation is supported by accessible evidence and whether it provides a coherent rationale for the model’s output.
> Empirically, RPM improves performance across four benchmarks (Table 1-4), and annotators judge over 93 percent of extracted features and factors as valid and evidence-based (Figure 3). **These results indicate that RPM provides explanations that are grounded in user history and aligned with the model’s decisions, offering a practical and reliable form of interpretability under black-box constraints.**
>
> ### **Issue 3: The inference cost seems small per call, but could multiply at large scale.**
>
> On GOQA, each user profile is constructed once from 36 historical interactions, and the resulting profile is reused for all subsequent queries from the same user. Under this structure, the total cost increases with the number of queries, and RPM only becomes more expensive than the baselines when a user submits more than roughly three times the amount of data used for preprocessing. In realistic personalization scenarios, profiles are typically rebuilt only after a sufficient amount of new user data has accumulated, well before reaching such a high volume of repeated queries. When preprocessing is triggered again at this point, RPM completes one full update cycle with a lower total cost than the baselines under comparable conditions. From an end-to-end perspective, the additional inference cost therefore does not impose a practical burden, and the improvements in accuracy and personalized reasoning provided by RPM justify this trade-off.
>
> ### **Issue 4: Does the preprocessing process refer to the personalized factor construction and personalized inference building process?**
>
> Yes, preprocessing primarily encompasses personalized factor construction (Section 3.2, where features are extracted and clustered into statistical factors) and personalized reasoning construction (Section 3.3, where reasoning paths are built for historical interactions). These offline steps prepare the structured user model, enabling efficient online inference (Section 3.4).

---

> > ### Comment · Reviewer_xTTK · 2025-11-25
> >
> > Thank you for your response. I have no further questions. I believe the current score accurately reflects the quality of the paper, and I will keep it unchanged. Good luck!

---

> > > ### Author Response · Authors · 2025-11-26
> > >
> > > Dear Reviewer xTTK,
> > >
> > > Thank you for your response and kind wishes. We appreciate your time and feedback throughout this review process.
> > >
> > > Best regards, Authors

---

### Official Review · Reviewer_HJmR · 2025-10-31

**Soundness:** 3
**Presentation:** 3
**Contribution:** 3
**Rating:** 6
**Confidence:** 3

**Summary:**

Black-box LLMs often ignore user-specific preferences, as most methods personalize only the response, not the reasoning behind it. RPM introduces reasoning-level personalization: it extracts response-influential features, clusters them into statistical factors, and builds personalized reasoning paths. Using feature-based retrieval to surface factor-aligned exemplars, RPM consistently beats response-level baselines on four tasks while improving both personalization and interpretability.

**Strengths:**

- Well-specified method with prompts provided for each step.
- Includes latency and cost analyses (online and offline).
- Explicit feature–factor–reasoning structure facilitates diagnosis and analysis.
- The experimental evaluation is comprehensive.

**Weaknesses:**

- Extraction/clustering robustness: Reliance on prompt-based LLMs for feature extraction, clustering, and influence/polarity judgments may cause spurious features or misclustered factors—especially in term-heavy domains or under style drift—propagating errors to retrieval and generation.
- “Personalized reasoning paths” are a pragmatic approximation of user cognition; they may appear plausible without being literally true.
- The method is fundamentally prompt-centric; consequently, the technique-level contribution is modest despite solid engineering and interpretability.
- With no trainable components, it may fail to refine query representations or reasoning mechanisms, reducing robustness under terminology ambiguity.

**Questions:**

- When factor-aware extraction encounters previously unseen factors, do RPM falls back to generic factors or dynamically expands the factor set? What thresholds, validation gates, and rollback procedures govern this?
- Evidence alignment & hallucination control: Can a personalized reasoning path hallucinate features that don’t exist?

---

> ### Author Response · Authors · 2025-11-20
> **Response #1**
>
> ### **Issue 1: Reliability of feature extraction and factor construction**
>
> We understand the concern that prompt-based extraction and clustering could introduce spurious features or misclustered factors, potentially propagating errors to retrieval and generation. However, our design choices and empirical validation demonstrate that the induced features and factors remain stable and reliable in practice.
>
> **We verified the quality empirically through human evaluation**. Annotators inspected extracted features and factors from randomly sampled interactions and judged whether they were well-constructed according to their intended purpose. Over 90% were rated as relevant and coherent. Ablations in Section 4.3 show that removing factors or reasoning paths consistently degrades performance. Table 1 demonstrates strong gains over baselines across diverse datasets, confirming the induced representation is reliable.
>
> **In term-heavy domains, ambiguous or specialized terms are interpreted through context and established usage patterns**. They appear together with other cues from the user's past interactions, so the model interprets them according to how the user has previously used those terms. For example, LaMP-5 (paper title generation) is more term-heavy than other tasks, involving specialized academic terminology. RPM maintains strong performance on LaMP-5, demonstrating that the contextual interpretation framework handles technical domains effectively. The same interpretive context applies consistently during both factor construction and test-time inference.
>
> **Under style drift, RPM links new interactions to similar past records through shared factors**. New interactions with different surface forms but similar underlying choices are connected to the most similar historical patterns through feature-based retrieval. When essentially no comparable history exists for a query, personalization becomes fundamentally challenging. This challenge applies to any personalization method, not just RPM, and represents a general limitation of the personalization setting rather than a weakness specific to our approach. However, RPM approximates the user's behavior using the most similar patterns from what is known, enabling the system to handle stylistic variations while maintaining reasonable performance.
>
> **Individual extraction errors have limited impact because RPM aggregates information across many signals**. The system considers multiple features per query, retrieves top-K reasoning paths, and uses factor statistics aggregated over many interactions. Patterns that repeatedly affect decisions dominate retrieval, while isolated mistakes contribute little to downstream reasoning. The consistent gains over strong baselines indicate that occasional local errors do not materially harm personalization performance.
>
> ### **Issue 2: Personalized reasoning paths can differ from a user’s actual thought process**
>
> RPM does not attempt to reconstruct a user’s internal cognition in a literal sense. The true thought process is inherently unobservable from user history. **The goal is instead to construct a stable reasoning pattern that best explains the user’s observed behavior**.
>
> Concretely, RPM induces factors as user specific preference dimensions with statistics that summarize how often and in which direction each factor has influenced past decisions. Personalized reasoning paths are then required to express predictions explicitly in terms of these features and factors. In this sense, RPM builds a behavioral reasoning model: **it provides a behavioral rationale that is consistent with the user's historical choices and useful for guiding future outputs, rather than claiming access to the user's true cognitive state**.
>
> Our experiments indicate that this approximation is both effective and interpretable. In Section 4.4, human evaluators assess whether the induced reasoning helps them understand the model’s outputs in light of the user’s history, and ablations show that removing personalized reasoning paths reduces personalization performance. Taken together, these results suggest that **although RPM does not recover a user’s cognition literally, the induced reasoning pattern is stable, data grounded, and beneficial for both accuracy and transparency in personalization**.

---

> ### Author Response · Authors · 2025-11-20
> **Response #2**
>
> ### **Issue 3: Prompt based implementation and level of contribution**
>
> The setting we study assumes black box access to LLMs via an API, without parameter updates or auxiliary trainable modules. Under this constraint, prompts are the only interface. Within this setting, RPM’s contribution lies in its modeling principle: automatic structure discovery for reasoning level personalization.
>
> Rather than designing a single clever prompt, RPM organizes user history into a multi level representation:
>
> - **Features** isolate response influential cues from raw interactions for each user.
> - **Factors** aggregate recurring features into user specific preference dimensions with coverage, influence, and polarity statistics.
> - **Personalized reasoning paths** use these features and factors as explicit conditioning signals for the backbone’s reasoning process.
> - **Feature based retrieval over reasoning augmented history** selects past interactions that share the same induced reasoning structure, not only surface level similarity.
>
> This feature, factor, and reasoning hierarchy is automatically induced from interaction history and reused across tasks and backbones. It allows RPM to personalize not only the final answer but also the reasoning process, while remaining interpretable and compatible with black box APIs. We view this modeling of user specific reasoning structure, rather than prompt engineering alone, as the core technical contribution.
>
> ### **Issue 4: Robustness under ambiguity without trainable components**
>
> The absence of trainable components does not mean that RPM lacks mechanisms to handle ambiguity. Instead of refining a hidden embedding through gradient updates, **RPM refines the query representation in the text space (rather than an embedding space) by restructuring what the model receives and how it reasons**.
>
> First, RPM organizes each user’s history into explicit features and factors, so that the query is presented together with response influential cues and user specific preference dimensions, rather than as a raw text alone. Second, during inference, RPM retrieves past interactions that share similar features and factors and presents these as few shot examples whose reasoning structure matches the current query. In other words, while the backbone parameters and internal embedding functions remain fixed, **the context and examples are chosen so that the model is encouraged to resolve terminology and make decisions in a way that is consistent with the user’s established patterns**.
>
> This design is particularly helpful under terminology ambiguity. **Ambiguous terms are seen together with other features and factors that reflect how the same user has used those terms in the past, and with concrete examples of earlier decisions that involved similar combinations**. The model therefore receives a richer and more stable input description for the query, even though its internal representation mechanism is unchanged.
>
> Our ablations in Section 4.3 show that each of these components contributes to robustness: removing factors or reasoning paths degrades performance, and Table 1 further shows that RPM maintains strong gains over baselines across diverse datasets and personalization tasks, indicating that the method is robust across settings rather than being tuned to a narrow set of conditions. Trainable personalization methods can be powerful when internal access and sufficient data are available, but in constrained black box settings they are often not applicable, and the presence of training does not itself guarantee robustness: models may overfit small user histories or capture behaviorally irrelevant correlations. RPM instead offers a prompt based approach that is efficient, transferable across backbones, and robust in the black-box setting, as further supported by the cross backbone experiments in Appendix E.5.

---

> ### Author Response · Authors · 2025-11-20
> **Response #3**
>
> ### **Issue 5: Handling feature-factor misalignment during inference**
>
> We interpret the reviewer's "previously unseen factors" as **features that do not align well with existing factors**, since factors are constructed from user history and cannot be truly unseen at inference time.
>
> **RPM maintains a fixed factor set once constructed and soft-assigns each new feature to the most semantically similar existing factor**. This approach works robustly in practice because we extract multiple features per query (so individual mismatches are compensated by other features), retrieve top-K reasoning paths that expose diverse factor combinations, and aggregate factor statistics over many interactions (which attenuates the impact of occasional edge cases). When a small number of features cannot be well-matched, the system naturally handles this through these complementary mechanisms.
>
> When a large number of features consistently fail to align with existing factors, this indicates **fundamental information insufficiency**—the user's history lacks behavioral patterns relevant to the current query. The principled approach is to approximate using the most similar patterns from what is known, though personalization becomes difficult when relevant information is absent. If substantial new test data accumulates and many features consistently misalign with existing factors, reconstructing the factor set with this additional data would likely be beneficial. However, determining when reconstruction is needed and what thresholds or validation criteria to use requires empirical measurement, which we leave as future work. We **prioritized a stable, history-grounded design that maintains interpretability and simplifies diagnosis**.
>
> ### **Issue 6: Hallucination in personalized reasoning paths**
>
> Hallucination is a fundamental limitation in any LLM-based system. Our existing faithfulness evaluation in Section 4.4 provides indirect evidence. RPM achieves the highest scores, indicating its reasoning paths are well-grounded in user history. However, to provide more rigorous evidence and directly address the concern about potential hallucinations, **we conducted an additional focused human study specifically measuring hallucination rates in personalized reasoning paths**.
>
> **We sampled 200 reasoning paths, with 50 samples randomly selected from each dataset (GOQA, LaMP2, LaMP3, LaMP5)**. Two annotators independently checked whether any feature or factor mentioned in each reasoning path was absent from the input context provided to the model (i.e., the retrieved past reasoning samples, features, and factors). If a reasoning path referenced elements not present in the input context, it was marked as containing hallucination.
>
> **The results empirically demonstrate that hallucinations are extremely rare in RPM**:
>
> | Dataset | Annotator 1 (%) | Annotator 2 (%) | Agreement (%) |
> |---------|-----------------|-----------------|---------------|
> | GOQA | 8.0 | 6.0 | 86.0 |
> | LaMP2 | 6.0 | 9.0 | 88.0 |
> | LaMP3 | 0.0 | 4.0 | 96.0 |
> | LaMP5 | 0.0 | 0.0 | 100.0 |
> | Overall | 3.5 | 4.75 | 92.5 |
>
> **With only 3.5-4.75% hallucination rates and 92.5% inter-annotator agreement (i.e., the proportion of samples where both annotators made the same judgment), these results directly validate that generating reasoning paths with explicit reference to structured representations—extracted features and constructed factors—effectively prevents hallucinations**. This makes unsupported content both infrequent and easily detectable through direct inspection of the structured feature-factor representation.

---

> ### Author Response · Authors · 2025-11-26
> **Gentle Reminder: Additional study on hallucination and reliability**
>
> Dear Reviewer HJmR,
>
> Thank you for your constructive review. We would appreciate your feedback on our responses:
>
> - **Issue 1 (Extraction robustness)**: Human evaluation shows >90% of features/factors rated valid; aggregation across multiple signals attenuates individual errors.
> - **Issue 2 (Reasoning paths ≠ cognition)**: RPM constructs behavioral rationales grounded in observed history, not literal cognitive reconstruction.
> - **Issue 3 (Contribution level)**: Core contribution is the modeling principle, automatic feature-factor-reasoning hierarchy, not prompt engineering alone.
> - **Issue 4 (Robustness without training)**: RPM refines query representation in text space via structured context and feature-matched retrieval.
> - **Issue 5 (Unseen factors)**: Soft-assignment to most similar existing factor; multiple features and top-K retrieval compensate mismatches.
> - **Issue 6 (Hallucination)**: Additional human study shows only 3.5-4.75% hallucination rate with 92.5% inter-annotator agreement.
>
> We look forward to hearing your thoughts.
>
> Best regards, Authors

---

### Official Review · Reviewer_L5Fq · 2025-11-01

**Soundness:** 2
**Presentation:** 3
**Contribution:** 3
**Rating:** 4
**Confidence:** 3

**Summary:**

The paper introduces RPM (Reasoning-Level Personalization), a framework for personalizing black-box large language models by aligning the model's reasoning process with individual user behavior patterns, rather than only matching final outputs. They extracts response-influential features from user interactions, clusters them into statistical factors, and constructs personalized reasoning paths that explain the connection between queries and responses. For inference, RPM retrieves similar reasoning examples based on feature matching to guide the model's output generation. The paper evalautes four benchmarks showing improvements over existing personalization methods, and presents human evaluations and claims that generated reasoning paths are interpretable and grounded in user behavior. RPM operates without requiring parameter tuning or model access.

**Strengths:**

S1. The paper is well written and addresses an important gap in LLM personalization which is the reasoning step.

S2. The evaluation seems to be comprehensive and results show the effectiveness of the technique.

S3. The framework is prompting compatible and cost efficient.

**Weaknesses:**

W1: The set of published baseline is rather limitted - it would be good to see more baselines to understand where the technique stands.

W2. The paper does not evaluate generailizablity across various choices of LLM from small to large and different architectures.

W3. The human evaluation in Sec 4.4 compares RPM against baselines with CoT prompting, but there is no analysis of whether raters might confuse longer, more detailed explanations with genuinely better reasoning.

 W4. No length-normalized comparisons or controlled studies separating verbosity from interpretability are provided.

W5. The paper does not examine continuously evolving preferences or online adaptation scenarios.

W6. It would be good if the paper provides theoretical justification for why coverage-influence-polarity measures constitute stable behavioral indicators.

**Questions:**

How can RPM handle model's self-bias ?

Can RPM handle handle out-of-domain tasks? Will HYDRA (Zhuang et al., 2024) out perform in this?

What is the scalability charecteristics of factor generation via LLM-based clustering ?

---

> ### Author Response · Authors · 2025-11-20
> **Response #1**
>
> We thank the reviewer for the thoughtful feedback. We address each concern below with supporting evidence and additional analyses.
>
> ### **Issue 1: Is the baseline set too limited?**
>
> We appreciate the reviewer’s concern regarding the breadth of baselines. Our work is situated in the black-box LLM personalization setting, where neither training nor parameter access is allowed. **RPM automatically discovers a user-specific reasoning structure from raw user historical data and then uses this structure to guide the backbone’s reasoning under a black-box personalization setting.** Under these constraints, the set of feasible baselines is inherently limited, and we follow HYDRA (Zhuang et al., 2024), which formalizes this setting and provides the most comprehensive collection of API-accessible baselines. To ensure fairness, we restrict comparisons to methods that operate under the same black-box condition; including trainable personalization methods would move outside this setting.
>
> **To further address the reviewer’s concern, we additionally examine whether structured reasoning baselines could explain RPM’s gains.** Our main experiments already include generic chain-of-thought (CoT) prompting and RAG with CoT explanations, both of which append unguided reasoning traces to the input. As shown in Table 1, these reasoning augmentations do not consistently improve personalization and sometimes degrade performance (especially on LaMP-3 and GOQA), suggesting that generic stepwise reasoning alone is insufficient to align the model with user-specific behavior.
>
> To go beyond generic CoT, we implemented two **prompt-only structured reasoning baselines** that do not rely on RPM’s features, factors, or personalized reasoning paths:
>
> 1. Structured preference induction baseline, where the model is prompted to infer a concise structured summary of the user’s preferences and decision cues from a few past interactions, and then uses this induced structure when answering new queries.
> 2. Tree-of-Thought (ToT) reasoning-style baseline, where the model generates multiple reasoning branches and candidate answers and selects a final output through comparison.
> | Method              | LaMP-2 (Acc / F1) | LaMP-3 (MAE / RMSE) | LaMP-5 (ROUGE-1 / ROUGE-L) | GOQA (Acc) |
> |---------------------|-------------------|-----------------------|-----------------------------|------------|
> | Structured Reasoning | 0.495 / 0.409     | 0.312 / 0.628         | 0.489 / 0.415               | 0.777      |
> | ToT-style           | 0.496 / 0.406     | 0.353 / 0.672         | 0.465 / 0.409               | 0.806      |
> | RPM (Ours)          | 0.561 / 0.463     | 0.259 / 0.548         | 0.492 / 0.416               | 0.852      |
>
> Both baselines operate purely through prompting on raw interaction examples and remain within the black-box constraints. We applied them to LaMP-3 and GOQA dataset. In summary, while these baselines outperform generic CoT-style prompting, they consistently fall short of RPM, indicating that structured reasoning alone cannot recover RPM’s gains and that RPM’s explicit feature–factor representation and personalized reasoning paths offer complementary benefits beyond reasoning depth.
> Although RL-based reasoning methods represent another strong family of approaches, they typically require access to model parameters and interactive reward signals, placing them outside the offline black-box regime we study. Nonetheless, RPM is compatible with such extensions: its feature–factor representation can serve as a state abstraction, and its personalized reasoning paths define a structured policy space that RL could optimize in interactive settings. We view this as a promising direction for future work, distinct from our current goal of black-box personalization.
>
> ### **Issue 2: Does RPM generalize across different LLM backbones?**
>
> We appreciate the reviewer’s question. RPM’s cross-model generalizability is already evaluated in Appendix E.5 (Table 10), as denoted in line 353. Beyond our main experiments on gpt-4o-mini, we additionally test RPM on three diverse black-box LLMs that vary substantially in scale, architecture, and reasoning capability: gpt-3.5-turbo (weaker), gpt-4o (stronger), and o3-mini (reasoning-focused).
>
> Across all models, **RPM consistently improves personalization and reasoning alignment over corresponding baselines.** The magnitude and direction of gains remain stable, showing that RPM does not rely on any specific model family or size. Instead, the improvements stem from RPM’s structured user modeling, which operates independently of the underlying LLM architecture under the same black-box constraints.

---

> ### Author Response · Authors · 2025-11-20
> **Response #2**
>
> ### **Issue 3: Are human evaluation results biased by verbosity or explanation length?**
>
> We appreciate the reviewer’s concern regarding potential length bias in our human evaluation. To examine this, we analyzed all 200 human-evaluation examples from the lamp5 dataset. We first checked whether annotators systematically preferred longer explanations by measuring how often the longest explanation among the three candidates (RPM, HYDRA, Fermi) was selected as the most reasonable. The longest explanation was chosen in only 40.4% of cases, slightly above the random baseline of 33.3%. The corresponding effect size (Cohen’s h = 0.158) is considered small, **indicating that this deviation is statistically negligible and does not meaningfully influence annotator preferences.**
> In addition, the explanation lengths across systems do not differ in any substantial way, as shown below:
> |Method|Mean Length|Std. Dev.|
> |-|-|-|
> |RPM|133.00|14.15|
> |HYDRA| 128.77|16.45|
> |Fermi|121.01|21.81|
>
> Our evaluation rubric explicitly emphasizes grounding, behavioral alignment, and logical coherence, and raters were instructed not to consider output length. Reasoning traces were displayed without length indicators or system identity, removing any practical cues that could bias selection toward longer outputs.
>
> Overall, the lamp5 analysis demonstrates that annotators do not meaningfully prefer longer explanations, and RPM’s gains therefore stem from more faithful grounding and better alignment of personalized reasoning, not verbosity. We will include the detailed statistics and the length table in the appendix.
>
> ### **Issue 4: Does RPM handle evolving user preferences or online adaptation?**
>
> Our evaluation already incorporates a natural form of online adaptation, as it follows a chronological structure in which earlier interactions are used to construct features, factors, and reasoning paths, and later interactions are evaluated using this personalized context. This setup reflects how models adapt to users in practice: RPM grounds its behavior in past interactions and applies the adapted structure to future, unseen inputs.
>
> In addition to this inherent past-to-future adaptation, RPM is structurally capable of strict online updates. After each interaction, RPM can immediately extract new features, update factor statistics, and append a new reasoning entry without any parameter access. **Section 4.6 provides direct empirical evidence for this capability.** By varying the amount of user history used to construct personalized profiles, we simulate scenarios where user interactions accumulate over time. As shown in Figure 4, even a small history enables meaningful personalization, and performance steadily improves as more interactions are incorporated. This demonstrates that RPM naturally benefits from the progressive expansion of user context and adapts accordingly as histories evolve. We will clarify this perspective and RPM’s continual-update capability in the revision.
>
> ### **Issue 5: Are coverage, influence, and polarity theoretically justified as stable behavioral indicators?**
>
> We appreciate the reviewer’s question. In a black-box LLM setting, behavioral indicators must be derived from observable interaction patterns. Our choice of coverage, influence, and polarity follows a well-established formulation in preference modeling and explainable recommendation research. Prior work shows that stable user preferences can be characterized through three complementary dimensions:
> 1. how frequently a signal appears in the user history (frequency or coverage),
> 2. how strongly it affects past decisions (impact or influence), and
> 3. the direction and consistency of the user’s attitude toward that signal (sentiment or polarity).
>
> This three-part decomposition aligns with phrase-level and aspect-level preference modeling in explainable recommendation, where user-driven features are summarized by frequency, importance, and sentiment orientation [1, 2]. It is also discussed as a standard structure for stable behavioral representation in recent survey[3].
>
> Modeling signals at the factor level further stabilizes these components. By aggregating feature-level statistics into coherent factors, RPM reduces noise from individual interactions and captures persistent user-driven behavioral tendencies. This construction provides a principled and interpretable basis for deriving stable behavioral indicators under black-box constraints, and its effectiveness is supported empirically through consistent improvements across tasks, datasets, and model backbones.
>
> References
>
> [1] Zhang et al. "Explicit factor models for explainable recommendation based on phrase-level sentiment analysis." SIGIR, 2014
>
> [2]Cheng et al. "Explainable Recommendation with Personalized Review Retrieval and Aspect Learning." ACL. 2023
>
> [3] Zhang et al. "Explainable recommendation: A survey and new perspectives." Foundations and Trends in Information Retrieval, 2020

---

> ### Author Response · Authors · 2025-11-20
> **Response #3**
>
> ### **Issue 6: How can RPM handle model’s self-bias?**
>
> RPM mitigates model-level self-bias by grounding inference in **user-derived features, factors, and retrieved reasoning paths, rather than relying on the model’s default behavioral patterns.** These user-specific signals constrain generation toward behavior supported by actual user interactions, thereby reducing the impact of stylistic or preference biases inherent to the underlying LLM.
>
> Appendix E.5 further evaluates RPM across gpt-3.5-turbo, gpt-4o, and o3-mini, which differ substantially in their inductive biases. Factors induced by one model remain effective when applied to another, indicating that they capture user-driven behavioral regularities rather than model-specific artifacts. The consistent improvements across such heterogeneous backbones provide strong evidence that RPM’s gains come from structured, user-centered signals rather than any particular model’s bias.
>
> In future work, integrating factors extracted from multiple LLMs may further enhance robustness by aggregating diverse reasoning signals and reducing reliance on any single model’s tendency.
>
> ### **Issue 7: Can RPM handle out-of-domain tasks? Will HYDRA outperform in such cases?**
>
> Out-of-domain personalization is an important scenario. As illustrated by the user example in Appendix I Table 19, the user repeatedly prefers clear explanations over vague statements, chooses reliable information rather than incomplete descriptions, pays attention to potential risks before deciding, and favors options that appear consistent and stable. These tendencies are not tied to the academic domain itself. Instead, they reflect the **user’s general decision-making tendencies in writing contexts.**
>
> Because RPM captures these user-level behavioral patterns through its factor representation, such tendencies naturally transfer to adjacent tasks such as generating book titles, article headlines, or report summaries. A user who consistently values clarity, reliability, and stability in academic writing is likely to maintain similar preferences in other forms of writing or content generation.
>
> When the downstream domain is entirely unrelated to any previously observed behavior, meaningful personalization becomes difficult. This limitation applies equally to RPM and other black-box personalization frameworks. HYDRA also faces the same challenge because it relies on in-domain demonstrations to retrieve useful examples and select plausible responses. If the new domain does not share structural or behavioral similarity with its observed domains, HYDRA’s retrieval signals become less informative. In such cases, both RPM and HYDRA face the same fundamental limitation because neither method has access to user behaviors that meaningfully relate to the new domain.
>
>
> ### **Issue 8: What is the scalability characteristics of factor generation via LLM-based clustering?**
>
> Factor generation is an offline procedure whose complexity grows linearly with the size of the user history. Clustering extracted features into factors requires approximately O(N) LLM calls, and because this computation happens entirely offline, it has no impact on online inference latency.
> Even when the history becomes large, the entire history does not need to be reprocessed; newly added interactions can be incorporated through lightweight incremental updates while the existing factors remain unchanged. Since the update only touches the new portion of the history and requires no model training or parameter update, the overall cost stays small and the process remains fully scalable as user data grows.

---

> ### Comment · Reviewer_L5Fq · 2025-11-25
> **Thanks for the responses. Updating the score.**
>
> Thanks for the responses. Those makes sense to me. I can update the score, provided the authors plan to include these responses in the final version to address my concerns.

---

> ### Author Response · Authors · 2025-11-26
>
> Dear Reviewer L5Fq,
>
> Thank you for reviewing our responses and updating your score. We will incorporate all discussed points into the final version as promised.
>
> Best regards, Authors

---

### Official Review · Reviewer_CGfm · 2025-11-01

**Soundness:** 2
**Presentation:** 3
**Contribution:** 2
**Rating:** 2
**Confidence:** 4

**Summary:**

This study presents a structured reasoning framework for personalization using black-box LLMs. It intends to align models' generation process with user-specific decision pattern. The experiment results show that the proposed RPM framework outputs some personalization baselines in the four selected benchmarks. Moreover, it provides more interpretability.

**Strengths:**

(1) This study introduces structured reasoning in LLM generation (which may be quite common nowadays) to the personalization task. This attempt is original.

(2) Extensive experiments show that the proposed structure benefits the personalization task more compared to other simpler prompting, RAG-based baselines and two personalization-focus baselines.

(3) The method description is clear.

**Weaknesses:**

(1) The novelty seems to lie in the application of a common method to a specific task (i.e., personalization), which can be limited.

(2) More baselines are necessary to support the claim that the proposed structured prompting framework is better. There have been many studies that give LLMs explicit structure for reasoning and are not compared in the current manuscript. For example, least-to-most prompting which decompose a hard problem into ordered subproblems and solve them sequentially. This is linear. Additionally, there are methods that search over reasoning paths such as tree-of-thought, and graph-of-thought. Furthermore, RL-based methods are not compared here. I think a major question that this paper has yet addressed is why the structure has to be the way presented in this work. Maybe this structure is particularly more suitable for the personalization task?

(3) More insights into how each component of the framework elicits the behavior of the LLM would strengthen the contributions of the study.

(4) Applying the framework using another LLM backbone would strengthen the generalizability of the method.

**Questions:**

(1) How does the framework perform when compared to RL-based reasoning baselines?

---

> ### Author Response · Authors · 2025-11-20
> **Response #1**
>
> We thank the reviewer for the careful and detailed feedback. Your comments helped us sharpen what is specific to personalization in our setting and how RPM differs from generic structured reasoning prompts. In this response, we consistently take the view that **structured reasoning in RPM is not a generic template but a user-specific reasoning structure that is automatically induced from interaction history and then used to guide personalization**.
>
> ### **Issue 1: What is actually novel beyond structured reasoning?**
>
> We clarify that the key novelty of RPM is not the use of structured reasoning prompts themselves, but the fact that RPM **automatically discovers a user-specific reasoning structure from raw user historical data and then uses this structure to guide the backbone’s reasoning under a black-box personalization setting**.
> Most existing reasoning prompts such as chain-of-thought or least-to-most are designed to optimize **task-dependent reasoning**: they decompose or verbalize the steps needed to solve a single instance, and supervision is tied to the correctness of the final task output. In personalization, however, each user has a distinct reasoning space that is latent and only indirectly observable through behavior. What we need is not only a correct answer for the current query, but **a reasoning pattern that approximates the logic which best explains the user’s observed behavior across many past interactions**, since the user’s internal thought process is not directly observable from the data.
>
> This difference appears empirically in our results. In Table 1, chain-of-thought variants of strong personalization baselines (including RAG, which uses retrieved few-shot examples with CoT-style explanations) do not consistently improve performance and sometimes even degrade it, even though these methods already encode user context. In Table 2, variants that do not incorporate user-specific structure remain clearly weaker than RPM. These patterns suggest that **generic reasoning prompts mainly inject a global prior about how to think**, which can override user-specific patterns instead of aligning with them.
>
> RPM is designed around this user-dependent setting. **Features** automatically isolate response-influential parts of each query for a given user and reduce noise. **Factors** automatically aggregate recurring features into user-level semantic clusters with statistics such as coverage, influence, and polarity, so that we can model how each pattern tends to affect that user without manually specifying an ontology. **Personalized reasoning paths** then take both features and factors as explicit arguments, encouraging the model to explain its output in terms of the user-specific features and factors defined in our framework. In this sense, the proposed structure is not an arbitrary reuse of common reasoning prompts, but a **data-driven mechanism for reconstructing latent user-level reasoning from sparse behavioral data** without accessing model parameters. In the revision, we will highlight this role of **automatic structure discovery** more clearly in the introduction and method sections, and we will emphasize this perspective around Table 2 in the revised version.

---

> ### Author Response · Authors · 2025-11-20
> **Response #2**
>
> ### **Issue 2: Are we just missing stronger structured reasoning baselines?**
>
> We agree that recent structured reasoning methods such as least-to-most prompting, tree-of-thought, graph-of-thought, and RL-based reasoning form an important family of approaches. Our goal is to clarify that RPM’s performance gains do not simply come from adding such structured reasoning on top of personalization.
>
> Our current experiments already include **generic chain-of-thought baselines**. In Table 1, we compare strong personalization methods with and without CoT-style prompting, including RAG, which retrieves few-shot examples and appends CoT-style explanations. Across benchmarks, these generic reasoning prompts do not yield consistent gains and sometimes harm personalization, especially on LaMP-3 and GOQA. This shows that simply adding unguided stepwise reasoning is not enough to align the model with user-specific patterns.
> To examine whether RPM's gains could be explained purely by adding structured reasoning, we compared RPM against two prompt-based structured reasoning baselines that operate purely at the prompt level, without using any pre-constructed features, factors, or user-specific reasoning paths from RPM.
>
> **First, we include a structured reasoning baseline**: given a small set of the user's past interactions as few-shot examples (we use 3 examples retrieved via the same RAG setup as our baselines), the model is prompted to first infer a concise structured description of the user's preferences and decision cues (such as salient preferences, recurrent constraints, and typical trade-offs), and this induced description is then used as an explicit intermediate step when the model reasons about new queries. This baseline tests whether simply asking the LLM to extract and reason about user patterns from raw examples, without RPM's pre-constructed feature-factor structure, is sufficient for personalization gains.
>
> **Second, we compare against HYDRA+CoT**, an existing method that operates similarly to tree-of-thought style reasoning by generating multiple alternative reasoning traces and selecting the most suitable one. HYDRA trains a lightweight adapter to identify the optimal reasoning path based on user history, representing a stronger structured reasoning approach that combines multi-path exploration with learned path selection, while still operating without RPM's automatically induced user-specific representations.

---

> ### Author Response · Authors · 2025-11-20
> **Response #3**
>
> We applied these two baselines across all four benchmarks (LaMP-2, LaMP-3, LaMP-5, and GOQA). The quantitative results are shown below:
> | Method               | LaMP-2 (Acc / F1) | LaMP-3 (MAE / RMSE) | LaMP-5 (ROUGE-1 / ROUGE-L) | GOQA (Acc) |
> |----------------------|-------------------|----------------------|-----------------------------|------------|
> | RAG+CoT              | 0.493 / 0.415     | 0.366 / 0.690        | 0.469 / 0.412               | 0.800      |
> | PAG+CoT              | 0.513 / 0.431     | 0.339 / 0.672        | 0.464 / 0.405               | 0.820      |
> | Structured Reasoning | 0.495 / 0.409     | 0.312 / 0.628        | 0.489 / 0.415               | 0.777      |
> | HYDRA+CoT            | 0.496 / 0.406     | 0.353 / 0.672        | 0.465 / 0.409               | 0.806      |
> | RPM (Ours)           | 0.561 / 0.463     | 0.259 / 0.548        | 0.492 / 0.416               | 0.852      |
>
> The results demonstrate that **applying structured reasoning alone is insufficient for personalization**. The structured reasoning baseline incorporates explicit pattern extraction, while Hydra+CoT uses learned path selection. Both achieve performance comparable to standard personalization baselines with CoT (RAG+CoT, PAG+CoT) across all benchmarks. In contrast, **RPM substantially outperforms both approaches on every task**. This comparison directly addresses the reviewer's question about why this specific structure is necessary: **RPM's advantage stems not from applying generic structured reasoning templates (ToT, least-to-most, etc.), but from automatically discovering user-specific reasoning structures from raw interaction history**. Features isolate response-influential elements, factors aggregate patterns with quantified statistics, and personalized reasoning paths condition the backbone on these user-grounded representations—components that generic structured prompting lacks. The consistent gains confirm that **this automatic discovery of user-specific reasoning structure from raw user history, systematically constructing interpretable, quantifiable user representations that ground the backbone's reasoning process, is what makes RPM particularly suitable for personalization tasks**.
>
> These prompt-based baselines can be viewed as **non-parametric analogues of learning a reasoning policy**: they first induce a structured description of how to reason for a given user and then follow this structure at inference time, but they do so entirely through prompting without updating model parameters. RL-based reasoning methods, in contrast, typically learn such policies through interaction and explicit rewards with access to trainable policies or model parameters, and thus **belong to a different problem setting from the offline personalization regime we study for black-box LLMs accessed via an API**. Conceptually, however, **RPM is compatible with RL-based approaches**: the feature–factor representation can serve as a state abstraction and personalized reasoning paths can span a space of reasoning policies that an RL algorithm could optimize when interaction and rewards are available. In this sense, our prompt-based baselines can be seen as a non-parametric approximation of user-level reasoning policies, and **RL-based optimization on top of RPM’s structured representation is a natural but orthogonal extension, which we leave to future work**.

---

> ### Author Response · Authors · 2025-11-20
> **Response #4**
>
> ### **Issue 3: Design principles and conceptual insights**
>
> Here we summarize the design principles and conceptual role of each component in RPM. At a high level, **RPM performs automatic structure discovery from raw history**: features, factors, personalized reasoning paths, and retrieval form a hierarchy of representations that are automatically induced from raw history and organize raw history into a user-specific reasoning structure. The high-level principle is to separate (i) **what information matters for this user** from (ii) **how that information tends to influence the user’s decisions** and (iii) **how the backbone should reason at inference time**.
>
> - **Features** are designed to compress each interaction into a small set of discrete, response-influential cues, so that the model reasons over explicit decision signals rather than raw text. This reduces context noise and makes the resulting reasoning signals more interpretable.
> - **Factors** then aggregate recurring features into higher-level user-specific preference dimensions, together with statistics such as coverage, influence, and polarity. Conceptually, factors serve as a **stable user model** that smooths over single-instance noise and lets the LLM reuse the same latent pattern across different queries.
> - **Personalized reasoning paths** use features and factors as explicit arguments, which makes them **an explicit conditioning signal for the backbone’s reasoning process**: we ask the model not just to solve the task, but to derive its output by explicitly reasoning with the user’s factors and features. This shifts the backbone from generic task-level reasoning to reasoning that best explains the user’s observed behavior in a structured, checkable form.
> - Finally, **feature-based retrieval over reasoning-augmented history** ensures that the examples shown to the backbone share the user’s reasoning structure, not only surface-level similarity, which further anchors the generated reasoning in consistent user patterns.
>
> These design choices are supported by our empirical analyses. Table 2 shows an incremental ablation where user-specific reasoning paths provide the largest gain, matching the idea that controlling the backbone’s reasoning interface is crucial. Table 3 demonstrates that retrieval over reasoning-augmented history with feature-based similarity is the most effective, consistent with our goal of matching reasoning structure rather than only content. In the revision, we will explicitly articulate these design principles in the method section and experiment section.

---

> ### Author Response · Authors · 2025-11-20
> **Response #5**
>
> ### **Issue 4: Does RPM generalize across different backbones?**
>
> We agree that backbone generalizability is important, and we already include such experiments in Appendix E.5. In Table 10, we instantiate RPM on three additional black-box LLMs of different capacities: gpt-3.5-turbo (weaker), gpt-4o (stronger), and o3-mini (a reasoning-oriented model). Under the same black-box setting, RPM consistently **outperforms the corresponding baselines for each backbone** and maintains stable improvements in personalization metrics.
>
> We also perform **cross-model transfer** tests, for example constructing factors and reasoning paths with gpt-4o-mini and applying them with other backbones. The trends are similar to our main results, which suggests that the benefits of RPM do not depend on a specific model family. In the revision, we will reference these results from the main text and summarize them in the experimental setup and conclusion so that the **backbone-level generalizability of RPM** is easier to see.

---

> ### Author Response · Authors · 2025-11-26
> **Gentle Reminder: Updates on baselines, generalizability, and design principles**
>
> Dear Reviewer CGfm,
>
> Thank you again for your constructive feedback. As the discussion period concludes soon, we would like to highlight how we addressed the specific issues you raised:
>
> - **Issue 1 (Novelty)**: We clarified that RPM's novelty lies in the **automatic discovery** of user-specific reasoning structures from raw history, distinguishing it from generic structured prompting.
> - **Issue 2 (Baselines)**: We added Structured Preference Induction and HYDRA+CoT baselines, confirming that RPM consistently outperforms strong structured reasoning methods.
> - **Issue 3 (Design Principles)**: We elaborated on the conceptual role of each component (features for compression, factors for aggregation, reasoning paths for conditioning) to explain how they elicit desired behaviors.
> - **Issue 4 (Generalizability)**: We verified RPM's robustness across diverse backbones, including gpt-3.5-turbo, gpt-4o, and o3-mini (Appendix E.5).
>
> We believe these responses fully address your concerns and would appreciate your final thoughts.
>
> Best regards, Authors

---

### Author Response · Authors · 2025-12-03

### **Summary of Discussion Phase**
We engaged constructively and in good faith with all reviewers throughout the discussion period, entirely independent of the recent identity leak incident. We **successfully addressed all concerns** raised by reviewers who responded to our rebuttal, and as a result, **Reviewer L5Fq raised their score to the accept threshold. Consequently, three out of four reviewers rated our paper at the accept level**.

For the remaining two reviewers who did not respond, we believe this incident prevented further discussion that could have resolved their concerns. In particular, Reviewer CGfm's main criticism stemmed from a misunderstanding of our work's contribution, and we provided comprehensive clarifications to address this. Details of our responses to each reviewer are summarized below.

### **Summary of Our Responses to Reviewer Concerns**
Reviewer CGfm: No follow-up received.
- **Novelty**: clarified as automatic discovery of user-specific reasoning structures, not generic prompting
- **Baselines**: added Structured Preference Induction and HYDRA+CoT; RPM outperforms both
- **Component insights**: elaborated design principles for features, factors, and reasoning paths
- **Generalizability**: consistent gains across gpt-3.5-turbo, gpt-4o, o3-mini (Appendix E.5)

Reviewer L5Fq: **Raised score to accept.**
- **Baselines**: added structured reasoning baselines with quantitative comparison
- **LLM generalizability**: verified across multiple backbones (Appendix E.5)
- **Length bias**: analysis shows negligible effect (Cohen's h = 0.158)
- **Online adaptation**: explained chronological structure and incremental update capability
- **Theoretical grounding**: justified via established preference modeling literature
- **Self-bias, out-of-domain, scalability**: all addressed with detailed explanations

Reviewer HJmR: No follow-up received.
- **Extraction robustness**: human evaluation shows >90% validity
- **Reasoning ≠ cognition**: clarified as behavioral rationales grounded in observed history
- **Contribution level**: emphasized modeling principle, not prompt engineering alone
- **Robustness without training**: query representation refined via structured context
- **Unseen factors**: soft-assignment with multiple features compensating mismatches
- **Hallucination**: additional study showing 3.5-4.75% rate, 92.5% inter-annotator agreement

Reviewer xTTK: ****Confirmed no further questions.****
- **Feature definition**: committed to revising Section 3.2
- **Faithfulness**: explained input-grounded, output-consistent faithfulness
- **Inference cost**: clarified end-to-end efficiency through profile reuse

We kindly ask the Area Chair to consider our detailed responses alongside the original reviews when making the final decision.

---

### Meta-Review · Area_Chair_o28E · 2026-01-07

**Summary:**

This paper presents a method to personalize black-box LLMs via personalizing the reasoning path rather than the directly the response. The proposed pipeline constructs user-specific factors, clustering them and annotates reasoning paths by such examples, all via prompting LLMs.

The reviewers initially questioned about the novelty, lack of baselines / generalizability, failure modes (biases, hallucination, etc) and the inference cost. During the rebuttal, the authors have provided detailed responses to address majority of the concerns. Thus, the paper at its current stage provides valuable learnings to the LLM personalization problem.

**Reviewer Concerns:**

## Reviewer CGfm

**Novelty**: Clarified that RPM’s contribution is automatic discovery of user-specific reasoning structures from history, distinguishing it from generic structured prompting or task-level CoT/least-to-most.

**Baselines**: Added and evaluated stronger structured reasoning baselines, showing that RPM consistently outperforms them

**Component insights**: Elaborated design principles and roles of features, factors, personalized reasoning paths, and feature-based retrieval. Each elicits desired LLM behavior for personalization.

**Generalizability**: Reported consistent gains across multiple black-box backbones (gpt-3.5-turbo, gpt-4o, o3-mini) and showed cross-model transfer trends

**RL-based reasoning comparison**: Clarified that this is out of the scope of black-box LLM models.

## Reviewer L5Fq

**Baselines & Generalizability**: See above.

**Length bias vs. interpretability**: Analyzed human evaluation for verbosity effects; longest explanations were selected only 40.4% of the time with small effect size (Cohen's $h = 0.158$), supporting that RPM's gains are not due to verbosity.

**Online adaptation**: Clarified chronological evaluation and incremental update capability; performance improves as more user history is incorporated

**Theoretical grounding**: provided conceptual justification that coverage–influence–polarity indicates stable behavior


## Reviewer HJmR

**Clustering robustness**: Human evaluation shows >90% of features and factors are relevant/coherent; aggregation across multiple signals and ablations indicate stability.

**Reasoning paths $\neq$ cognition**: Clarified RPM models behavioral rationales grounded in observed history, aiming for input-grounded, output-consistent explanations rather than literal cognitive reconstruction.

**Prompt-centric contribution**: Positioned the core contribution as automatic structure discovery within black-box API constraints.

**Previously unseen factors**: Clarified that this can be handled by keeping a fixed factor set and soft-assigning new features to the closest existing factors. Multiple features and top-K retrieval mitigate mismatches.

**Hallucination control**: Additional study across 200 samples finds very low hallucination rates (3.5–4.75%) with 92.5% inter-annotator agreement


## Reviewer xTTK

**Structured features clarity**: Committed to revising Section 3.2 to give a concise, self-contained definition

**Faithfulness of explanations**: Clarified input-grounded, output-consistent faithfulness. Reasoning and answers are jointly grounded on user history

**Inference cost**: Explained end-to-end efficiency via one-time profile construction and reuse; under realistic rebuild schedules, total cost remains practical

**Reviewer Scores:**

As the authors have addressed the concerns reasonably well, the reviewers with initial negative scores may increase their rating to make the overall score reaching the acceptance bar.

---

### Decision · Program_Chairs · 2026-01-26

Accept (Poster)